# Thymoquinone as an electron transfer mediator to convert Type II photosensitizers to Type I photosensitizers

Jiahao Zhuang[1,2,5], Guobin Qi[2,5], Yecheng Feng[3], Min Wu[1], Hang Zhang [4], Dandan Wang[1,2], Xianhe Zhang [2], Kok Chan Chong[2], Bowen Li[2], Shitai Liu[1,2], Jianwu Tian[2], Yi Shan[2], Duo Mao[3] ✉ & Bin Liu [1,2] ✉

The development of Type I photosensitizers (PSs) is of great importance due to the inherent hypoxic intolerance of photodynamic therapy (PDT) in the hypoxic microenvironment. Compared to Type II PSs, Type I PSs are less reported due to the absence of a general molecular design strategy. Herein, we report that the combination of typical Type II PS and natural substrate carvacrol (CA) can significantly facilitate the Type I pathway to efficiently generate superoxide radical ($O_2^{-•}$). Detailed mechanism study suggests that CA is activated into thymoquinone (TQ) by local singlet oxygen generated from the PS upon light irradiation. With TQ as an efficient electron transfer mediator, it promotes the conversion of $O_2$ to $O_2^{-•}$ by PS via electron transfer-based Type I pathway. Notably, three classical Type II PSs are employed to demonstrate the universality of the proposed approach. The Type I PDT against *S. aureus* has been demonstrated under hypoxic conditions in vitro. Furthermore, this coupled photodynamic agent exhibits significant bactericidal activity with an antibacterial rate of 99.6% for the bacterial-infection female mice in the in vivo experiments. Here, we show a simple, effective, and universal method to endow traditional Type II PSs with hypoxic tolerance.

Photodynamic therapy (PDT) has been widely explored for combating a range of diseases[1–4]. Specifically, upon light irradiation, photosensitizers (PSs) are excited to generate reactive oxygen species (ROS) to induce cytotoxicity and cell inactivation[5]. Currently, the dominant mechanism of PSs is based on Type II pathway, which involves direct energy transfer from excited PS to molecular oxygen ($O_2$) to produce singlet oxygen ($^1O_2$) and is dependent on oxygen availability[5–10]. Unfortunately, hypoxic microenvironment is often presented in solid tumors and bacterial-infected tissues[11–15], which compromises the therapeutic efficacy of PDT. Recent studies have demonstrated that Type I PDT can perform well under hypoxic microenvironment[16]. In the

Type I pathway, electron transfer occurs when the PS first extracts an electron from nearby substrate and then transfers the electron to the surrounding oxygen, yielding radical species such as hydroxyl radicals (OH•) and superoxide radicals ($O_2^{-•}$). In contrast to the Type II pathway, Type I PDT demonstrates reduced reliance on oxygen, facilitated by the involvement of $O_2^{-•}$ species in disproportionation reaction, along with the Fenton and Haber−Weiss reactions. These processes not only regenerate oxygen to alleviate hypoxia but also stimulate the formation of other highly toxic ROS species[17,18]. As a result, Type I PDT shows great potential for antibacterial treatment within hypoxic microenvironment.

[1]Joint School of National University of Singapore and Tianjin University, International Campus of Tianjin University, Binhai New City, Fuzhou, China. [2]Department of Chemical and Biomolecular Engineering, National University of Singapore, Singapore, Singapore. [3]Institute of Precision Medicine, The First Affiliated Hospital of Sun Yat-Sen University, Sun Yat-Sen University, Guangzhou, China. [4]Department of Materials Science and Engineering, National University of Singapore, Singapore, Singapore. [5]These authors contributed equally: Jiahao Zhuang, Guobin Qi. ✉e-mail: maod6@mail.sysu.edu.cn; cheliub@nus.edu.sg

Currently, mature theoretical guidance for molecular design of Type II PSs has been developed[16,19,20]. In recent years, several strategies have been reported to design Type I PSs or endow Type II PSs with extra Type I ROS generation through cationization, heavy-atom regulation, and biotinylation of PSs[21–23]. However, a universal molecular design principle to guide the design of Type I PSs has yet to be established so far[24,25]. The difficulties lie in the fact that electron transfer process is not as efficient as energy transfer process, and both processes could happen in a competing relationship[26,27]. To promote the electron transfer process, sufficient close contact and appropriate redox potential among PS, substrate, and oxygen are required[26–30]. Therefore, introducing appropriate substrates with proper redox potential is a feasible strategy to facilitate the electron transfer process, thus promoting the generation of Type I ROS.

In this work, we report that the combination of natural substrate carvacrol (CA) and classical Type II PS (e.g., chlorin e6 as PS1) leads to the boosted generation of Type I $O_2^{-\cdot}$ through facilitated electron transfer process. We demonstrate that CA as a $^1O_2$ susceptible bioactive molecule can be in situ activated and then converted into thymoquinone (TQ) by local $^1O_2$ generated from PS upon light excitation (Fig. 1a). However, further studies discover that the generated intermediate TQ plays the role of an electron transfer mediator to facilitate electron transfer between PS and $O_2$, leading to the enhanced Type I pathway and effective generation of $O_2^{-\cdot}$ for relieving the restriction of hypoxic microenvironment at the lesion (Fig. 1b). In addition to PS1, two other PSs, namely PS2 and PS3, are employed to demonstrate the universality of the enhanced electron transfer process mediated by TQ in promoting the Type I pathway (Fig. 1c). Moreover, the Type II to Type I PDT antibacterial activity of TQ/PS1 complex has been successfully established under hypoxic conditions against *Staphylococcus aureus* (*S. aureus*) in vitro. Furthermore, the therapeutic performance of the TQ/PS1 complex is further confirmed in the bacterial-infection female mice model with a bactericidal rate of 99.6%. This work provides a simple, effective, and universal approach to convert Type II PS into Type I PS through an enhanced electron transfer process, endowing Type II PS with improved tolerance under a hypoxic microenvironment.

## Results

### Assessment of CA/PS1 complex

Initially, we found that the combination of Type II PS1 and natural substrate CA tends to form carrier-free complex CA/PS1 via small molecule self-assembly behavior (Supplementary Fig. 1). The formation of a carrier-free complex between PS1 and hydrophobic drug is through π-π stacking and hydrophobic interaction[31–33]. The UV–Vis absorbance of this complex exhibited characteristic peaks from PS1 (405 nm) and CA (273 nm), suggesting the co-existence of both molecules (Supplementary Fig. 2). Scanning electron microscope (SEM) image and dynamic light scattering (DLS) measurement showed that the complex was uniformly distributed in a spherical shape with an estimated hydrodynamic particle diameter of 111.7 ± 21.2 nm (Supplementary Fig. 3). Moreover, CA/PS1 complex exhibited good colloidal stability in phosphate-buffered saline (PBS) after 7-day storage at 4 °C (Supplementary Fig. 4).

Next, we investigated the ROS generation of PS1 and CA/PS1 complex to understand the effect of CA. A commercial overall ROS fluorescent indicator 2,7-dichlorodihydrofluorescein (DCFH), was employed to explore ROS generation. DCFH is non-fluorescent but exhibits significant green fluorescence upon oxidation by ROS. As shown in Fig. 2a, upon light irradiation (light spectrum is shown in Supplementary Fig. 5), the solution of DCFH in the presence of the CA/PS1 complex displayed about 433-fold fluorescence enhancement, which was amplified around 5.5-fold compared to that of PS1 alone, implying that CA can greatly promote ROS generation of PS1.

To understand more about different ROS generation, 9,10-anthracenediyl-bis(methylene)dimalonic acid (ABDA) was selected as a $^1O_2$ indicator. The $^1O_2$ generation in the CA/PS1 system was found to be dramatically reduced compared to that of PS1 alone (Fig. 2b), in accordance with the reported literature that CA can quench $^1O_2$[34,35]. This suggests that other species of ROS should be increased to account for the enhancement of overall ROS. Subsequently, dihydrorhodamine 123 (DHR 123), a commonly used indicator for $O_2^{-\cdot}$ detection, was used to detect the generation of $O_2^{-\cdot}$. PS1 as a porphyrin derivative is usually known as a classical Type II PS, which does not generate much $O_2^{-\cdot}$[23]. However, the fluorescence of DHR 123 solution was enhanced significantly when treated with CA/PS1 complex upon light irradiation

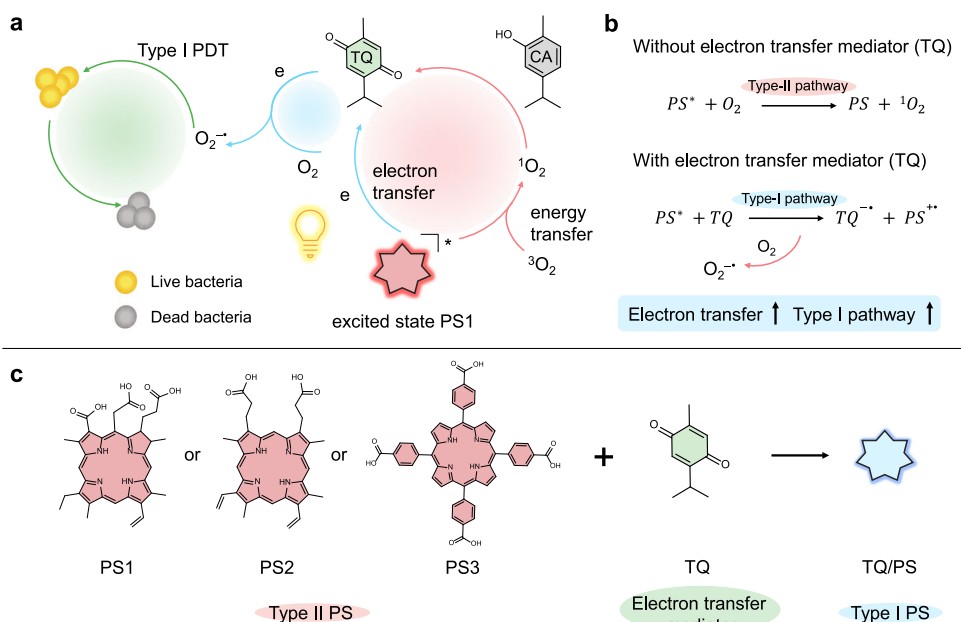

**Fig. 1 | Schematic illustration of the facilitated generation of $O_2^{-\cdot}$ through electron transfer strategy. a** Photoinduced conversion from CA to electron transfer mediator TQ and the subsequent electron transfer process for boosting the Type I pathway. **b** The PDT pathway in the absence or presence of electron transfer mediator TQ. **c** Chemical structures of Type II PSs and TQ.

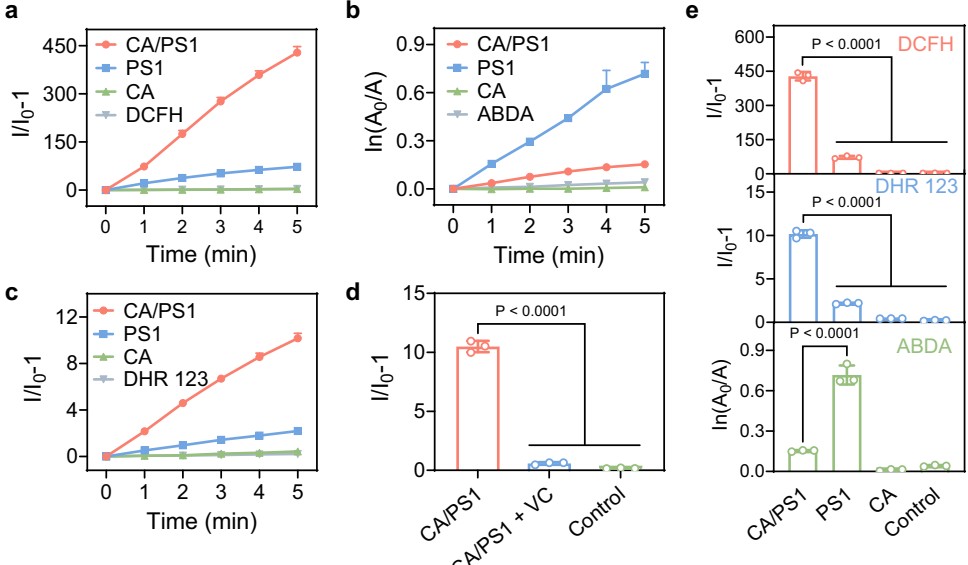

**Fig. 2 | ROS generation properties of CA/PS1 complex. a** Comparison of the PL intensity of DCFH alone and DCFH with CA/PS1 (PS1 4 μg/mL, CA 150 μg/mL), PS1 (4 μg/mL), and CA (150 μg/mL) under light irradiation for overall ROS detection. **b** Comparison of the decomposition of ABDA alone and ABDA with CA/PS1 (PS1 4 μg/mL, CA 1 mg/mL), PS1 (4 μg/mL), and CA (1 mg/mL) under light irradiation for $^1O_2$ detection. **c** Comparison of the PL intensity of DHR 123 alone and DHR 123 with CA/PS1 (PS1 4 μg/mL, CA 150 μg/mL), PS1 (4 μg/mL), and CA (150 μg/mL) under light irradiation for $O_2^{-\cdot}$ detection. **d** Comparison of the PL intensity of DHR 123 with or without radical scavenger under light irradiation. **e** Summary of different types of ROS generation in the presence of ROS detection probes alone and probes with different treatments. Data in (**a**–**e**) are presented as mean ± SD derived from $n = 3$ independent samples. Statistical significance was analyzed via one-way ANOVA test with a Tukey post hoc test.

(Fig. 2c). In addition, when vitamin C (VC) was introduced as a radical scavenger into the CA/PS1 solution, the fluorescence was quenched efficiently, which further confirmed that the enhanced DHR 123 fluorescence was attributed to the generation of $O_2^{-\cdot}$ (Fig. 2d). These results demonstrated that the generation of $O_2^{-\cdot}$ in CA/PS1 system is responsible for the enhanced generation of overall ROS despite reduced $^1O_2$ generation as summarized in Fig. 2e. It revealed that the combination of PS1 and CA endows the traditional Type II PS1 with unexpected Type I ROS generation property.

## In vitro bactericidal activities of CA/PS1 complex

Then, the CA/PS1 complex was used to evaluate the bactericidal activities towards the Gram-positive bacteria *S. aureus* as a representative model. The bactericidal effects of CA and PS1 alone were first explored, and both showed antimicrobial properties in a dose-dependent manner (Supplementary Figs. 6 and 7). Besides, there is no difference in bacterial uptake of PS1 regardless of the addition of CA (Supplementary Fig. 8). Then, bactericidal activities of CA, PS1, and the corresponding CA/PS1 complex were fully investigated using the colony-forming unit (CFU) plate counting method under identical conditions. Under normoxic conditions, the colony numbers of *S. aureus* dropped significantly after being treated with the CA/PS1 complex upon light irradiation (Fig. 3a and Supplementary Fig. 9), suggesting efficient photodynamic antibacterial activity of the CA/PS1 complex in vitro. In contrast, CA at 150 μg/mL can only slightly inhibit the growth of *S. aureus,* and 4 μg/mL PS1 showed a mild bactericidal effect. We also quantified the live cell number using CFU count (Fig. 3c). The group treated with the CA/PS1 complex exhibited a 5000-fold reduction in live cell number compared to those treated with PS1 alone. Furthermore, CA/PS1 complex was also found to exhibit efficient elimination capability against methicillin-resistant *Staphylococcus aureus* (MRSA) (Supplementary Fig. 10). These results demonstrated that the combination of PS1 and CA can efficiently boost antibacterial activities, which is in good agreement with the amplified ROS generation of CA/PS1 complex.

Moreover, inspired by the efficient generation of $O_2^{-\cdot}$ in CA/PS1 system and the hypoxic tolerance characteristic of Type I PDT, bactericidal activities under hypoxic conditions were further conducted. A notable increase in the number of live cells was observed in the PS1-treated culture (Fig. 3b and Supplementary Fig. 11), which is consistent with the compromised Type II PDT effect of PS1 under hypoxic conditions. On the other hand, CA/PS1 complex can still efficiently inactivate almost all bacteria with an antibacterial rate of 99.9998%, which is much better than the treatment by PS1 or CA alone. Specifically, according to the quantification of bactericidal effects under hypoxic conditions shown in Fig. 3d, the live cell counts ratio of treatment with PS1 compared to treatment with the CA/PS1 complex increased significantly with a remarkable 67,000-fold difference. This result demonstrated the effectiveness of the CA/PS1 complex in treating bacteria under hypoxic conditions. The importance of the generated $O_2^{-\cdot}$ in bactericidal activities was established by the addition of specific $O_2^{-\cdot}$ quencher VC (Supplementary Fig. 12). As expected, the bactericidal activity of CA/PS1 complex was inhibited by VC in a dose-dependent manner, indicating that Type I PDT is the main cause of death in bactericidal activities when treated with CA/PS1 complex.

The bactericidal activities were further confirmed by live and dead staining, in which the bacteria were co-stained with SYTO-9 and propidium iodide (PI) to differentiate between live and dead bacteria. Bacteria with intact cell membranes are stained with green fluorescence, whereas bacteria with damaged membranes are stained with red fluorescence. As depicted in Fig. 3e and Supplementary Fig. 13, almost all bacteria were stained by red color when treated with CA/PS1 complex under light irradiation, while much less red signal was observed in other control groups. Then, SEM was applied to analyze the morphological changes of *S. aureus* to assess the photodynamic ablation activities. Without light irradiation, *S. aureus* showed smooth and intact bodies after treatment by all groups (Fig. 3f). Upon light irradiation, CA/PS1 complex induced much more severe damage to bacteria compared to the blank and control groups, which was supported by the evident contraction and collapse of the bacterial membrane. Collectively, these results demonstrated that CA/PS1 complex

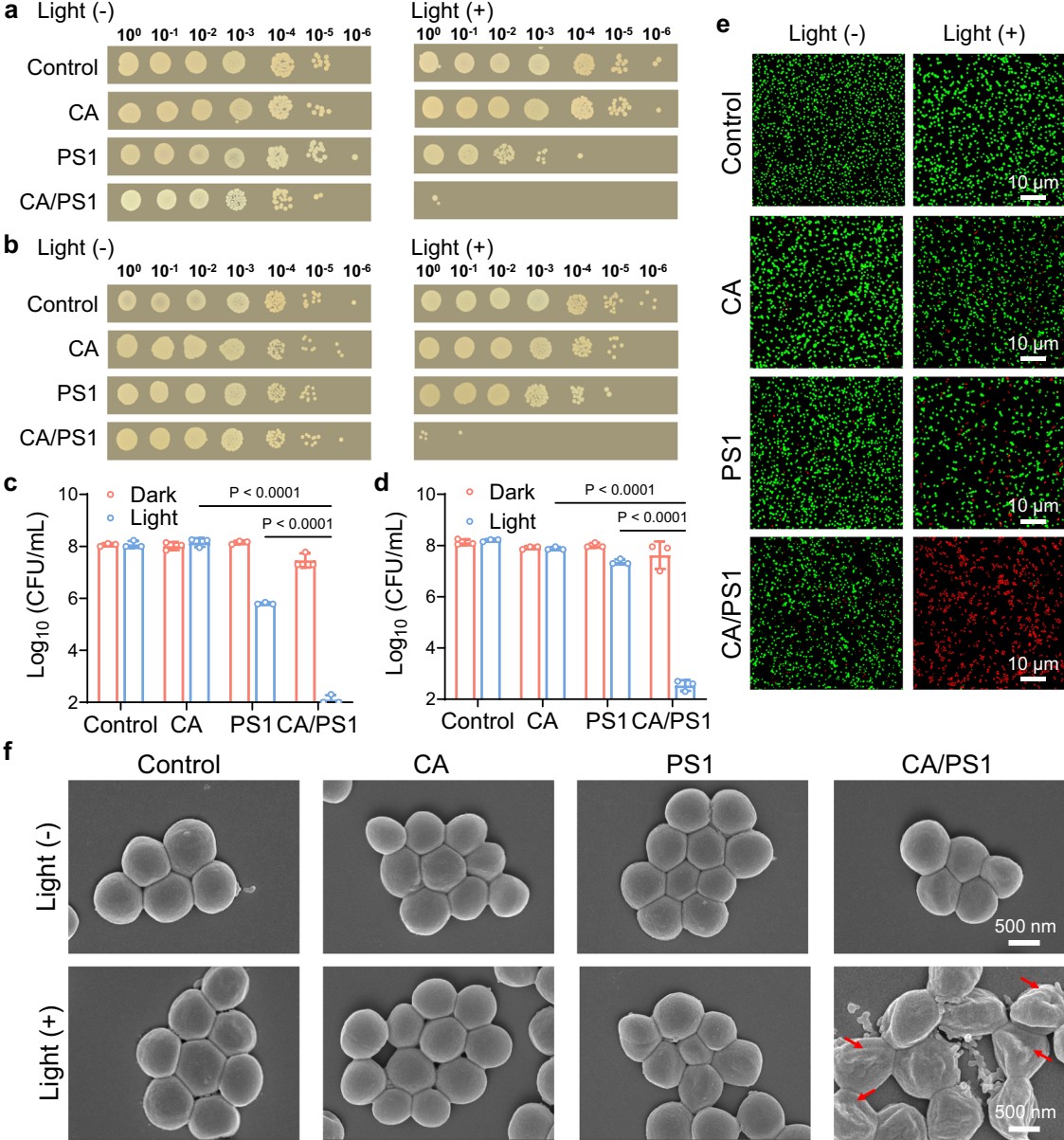

**Fig. 3 | Bactericidal activities of CA/PS1 complex.** Photographs of the LB agar plates after the inoculation and overnight incubation of *S. aureus* cultures when treated under normoxic (**a**) or hypoxic (**b**) conditions with or without light irradiation (60 mW/cm², 10 min) in different groups. **c, d** CFU counts from the antibacterial assay as shown in (**a**, **b**), respectively. Data in (**c**, **d**) are presented as mean ± SD derived from *n* = 3 independent biological samples. Statistical significance was analyzed via two-way ANOVA test with a Tukey post hoc test.

**e** Representative live/dead bacterial staining images of *S. aureus* treated with or without light irradiation in different groups. Green fluorescence: live bacteria, red fluorescence: dead bacteria. Scale bar = 10 μm. **f** Representative SEM images of *S. aureus* treated with or without light irradiation in different groups. Scale bar = 500 nm. Different groups include CA/PS1 (PS1 4 μg/mL, CA 150 μg/mL), PS1 (4 μg/mL), and CA (150 μg/mL).

with amplified ROS generation ability can be used as an efficient bactericidal agent, moreover, the facilitated $O_2^{-\cdot}$ generation in the complex makes the traditional Type II PS adaptive to hypoxic conditions.

**TQ as intermediates play an important role in CA/PS1 complex**
To understand the underlying mechanism of the unexpectedly enhanced production of $O_2^{-\cdot}$, gas chromatography-mass spectrometry (GC-MS) was applied to analyze the possible by-products of CA when mixed with PS1 upon light irradiation. In addition to the main peak of CA located at 2.47 min, the occurrence of a new peak at 2.28 min was observed after light irradiation, which was absent without light irradiation (Fig. 4a). The structure of this new compound was further verified by GC-MS analysis, indicating that the oxidation product of CA is TQ. This is in accordance with the standard reference of TQ with the

same retention time at 2.28 min (Supplementary Figs. 14 and 15). Indeed, it has been reported that the phenolic hydroxyl group of CA can undergo oxidation to form TQ with para-benzoquinone structure in the presence of $^1O_2$[34,36,37]. Given the generated TQ as the dominant intermediate product, we hypothesize that TQ may play an important role during the facilitated generation of $O_2^{-\cdot}$.

Due to their hydrophobic structures, PS1 and TQ were also found to self-assemble into a spherical shape, and the UV−Vis result confirmed the co-existence of PS1 and TQ inside TQ/PS1 complex (Supplementary Figs. 16 and 17). Then, the generation of $O_2^{-\cdot}$ from the TQ/PS1 complex was evaluated by utilizing DHR 123 as an indicator (Supplementary Fig. 18). Indeed, the fluorescence of DHR 123 solution was dramatically increased when treated with TQ/PS1 complex upon light irradiation. In contrast to TQ/PS1 complex, TQ itself did not

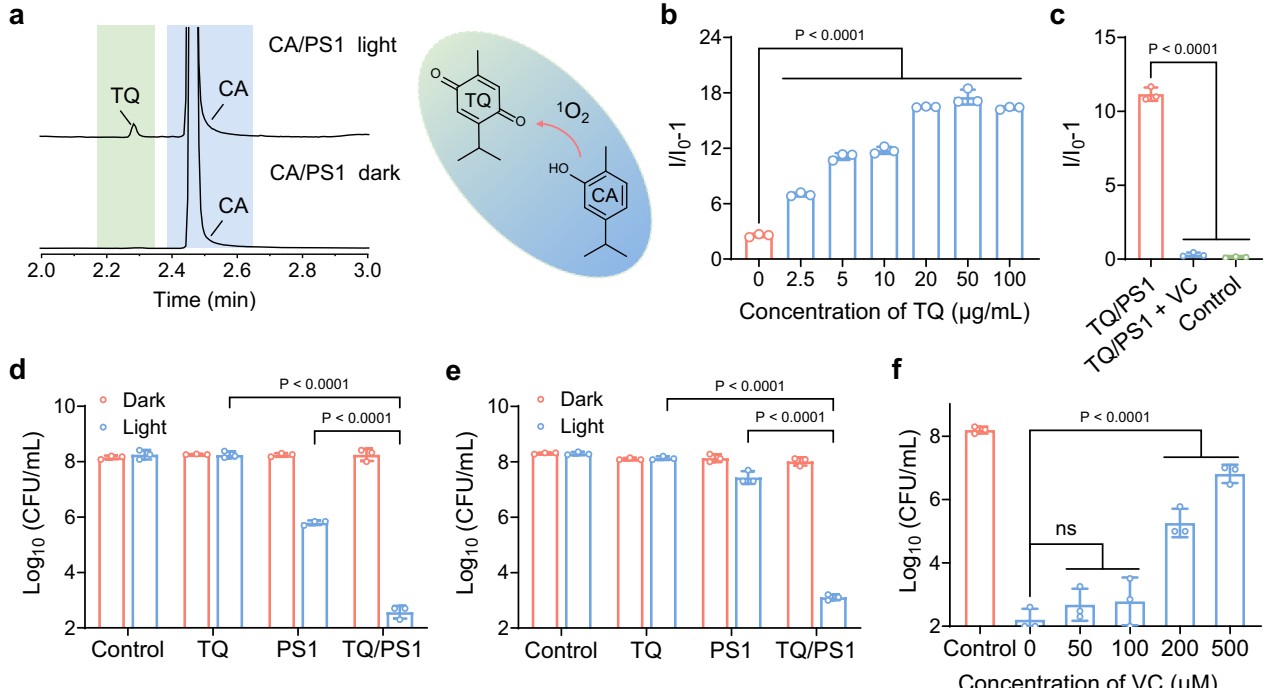

**Fig. 4 | ROS generation properties and bactericidal activities of TQ/PS1 complex. a** GC-MS analysis of photoinduced oxidization of CA in the presence of PS1 with or without light irradiation. **b** Comparison of the PL intensity of DHR 123 with the addition of different concentrations of TQ (0-100 µg/mL) in the presence of PS1 (4 µg/mL) under light irradiation. **c** Comparison of the PL intensity of DHR 123 with or without radical scavenger under light irradiation. CFU counts of *S. aureus* treated with TQ/PS1 (PS1 4 µg/mL, TQ 8 µg/mL), PS1 (4 µg/mL), or TQ (8 µg/mL) under normoxic (**d**) or hypoxic (**e**) conditions in the presence or absence of light irradiation (60 mW/cm², 10 min). **f** Abolishment of the antibacterial activities by the addition of $O_2^{-\cdot}$ scavenger VC in the presence of TQ/PS1 complex (PS1 4 µg/mL, TQ 8 µg/mL). Data in (**b**–**f**) are presented as mean ± SD derived from *n* = 3 independent samples. Statistical significance was analyzed via one-way (**b**, **c**, **f**) or two-way (**d**, **e**) ANOVA test with a Tukey post hoc test. ns no significance.

exhibit obvious production of $O_2^{-\cdot}$. Furthermore, the yield of $O_2^{-\cdot}$ by TQ/PS1 complex under different concentrations of TQ was compared (Fig. 4b), demonstrating that the amount of TQ at around 20 µg/mL would be an optimized ratio for efficient generation of $O_2^{-\cdot}$. Besides, VC was also introduced as a radical scavenger to further confirm the generation of $O_2^{-\cdot}$ with the fluorescence of DHR 123 solution was expectedly eliminated as positive evidence (Fig. 4c). These results showed that the boosted generation of $O_2^{-\cdot}$ in the CA/PS1 complex should be attributed to the presence of TQ.

Meanwhile, the bactericidal activities of TQ/PS1 complex were also investigated. The inherent bactericidal activity of TQ against *S. aureus* was negligible regardless of light irritation (Supplementary Fig. 19). Next, a notable diminution of colony numbers of *S. aureus* was observed upon treatment with TQ/PS1 complex under light irradiation (Supplementary Fig. 20). The quantification result demonstrated that the one treated with TQ/PS1 complex was much more efficient than those treated by PS1 alone (Fig. 4d), exhibiting a similar synergistic effect as CA/PS1 complex. Then, the bactericidal activity of TQ/PS1 under different concentrations of TQ was explored (Supplementary Fig. 21), exhibiting a dose-dependent manner and almost all bacteria were eliminated when the concentration of TQ reached 8 µg/mL. Moreover, the amount of TQ used was around 5% of CA for achieving a similar synergistic ablation of bacteria effect, in agreement with the fact that only a small portion of CA was converted to TQ when oxidized by $^1O_2$ (Fig. 4a). Motivated by the enhanced Type I ROS generation and efficient bactericidal activities exhibited by the TQ/PS1 complex, antibacterial experiments were conducted under hypoxic conditions to assess the Type I PDT of the TQ/PS1 complex. The results demonstrated that TQ/PS1 complex remains highly effective in eliminating bacteria under hypoxic conditions, in contrast to the scenario with PS1 alone in a Type II mode (Fig. 4e and Supplementary Fig. 22). Afterward,

VC was applied to further confirm that the bacteria were eliminated by Type I PDT through quenching of the generated $O_2^{-\cdot}$ from TQ/PS1 complex at bacterial level, with the decrease of bactericidal activities observed in a dose-dependent manner (Fig. 4f and Supplementary Fig. 23). Based on the above ROS evaluation and bactericidal results of TQ/PS1 complex, we suggest that the good performance of CA/PS1 complex was originated from the intermediate TQ.

**Boosted electron transfer process mediated by TQ**

A series of experiments were then conducted to understand the role of TQ and interaction with PS1 to achieve amplified ROS generation and Type I PDT. Considering the working principle of Type I PDT and the intrinsic electron-deficient property of TQ, we suspect that the promoted Type I pathway may be attributed to the boosted electron transfer process between PS1 and TQ. First, the steady-state fluorescence quenching of PS1 in the presence of TQ was explored. Obviously, the fluorescence of PS1 was quenched with the increasing concentration of TQ (Fig. 5a). The quenching constant ($K_q$) was determined using the Stern−Volmer equation:

$$I_0/I = 1 + K_{sv}[Q] \quad (1)$$

$$K_{sv} = K_q \times \tau \quad (2)$$

Where $I_0$ is the fluorescence intensity of PS1 in the absence of TQ; $I$ is the fluorescence intensity of PS1 in the presence of different concentrations of TQ; $\tau$ is the fluorescence lifetime of PS1 in DMF solution. The $K_q$ was calculated to be $1.92 \times 10^{10}\,M^{-1}\,S^{-1}$ (Fig. 5b), indicating that TQ can effectively quench the excited state of PS1 by electron transfer process[27]. In contrast, CA by itself cannot quench the emission of PS1,

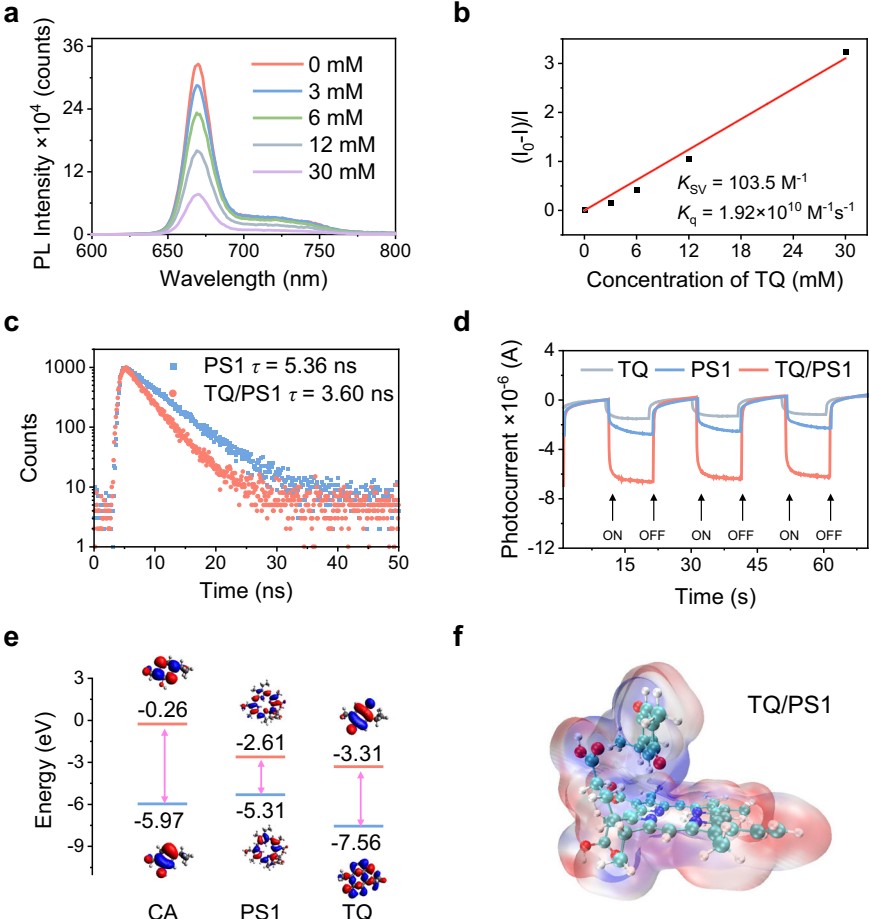

**Fig. 5 | The study of electron transfer between PS and TQ. a** The emission spectra change of PS1 (10 μM) with increasing concentrations of TQ (0–30 mM) under excitation at 405 nm in DMF. **b** Stern–Volmer plot of fluorescence intensity change of PS1 against TQ in DMF. **c** Fluorescence decay curves of PS1 and TQ/PS1 at a peak of 670 nm. **d** Photocurrent responses of PS1, TQ, and TQ/PS1 systems. **e** HOMO and LUMO energy levels of CA, PS1, and TQ. **f** Simulated electrostatic potential map of TQ/PS1.

and the fluorescence of PS1 was only gradually quenched when CA was treated by light irradiation (Supplementary Figs. 24 and 25), suggesting that the electron transfer process was promoted by the intermediate product TQ instead. Then, the fluorescence lifetime of PS1 with the addition of TQ was studied. As shown in Fig. 5c, the fluorescence lifetime of PS1 (5.36 ns) was shortened to 3.60 ns in the presence of TQ, further supporting the electron transfer process between PS1 and TQ. Thus, TQ is expected to act as a simple and effective mediator to boost the electron transfer process for facilitating Type I pathway.

On the other hand, the photocurrent responses of PS1, TQ, and TQ/PS1 systems were investigated to study their charge separation abilities. As shown in Fig. 5d, the TQ/PS1 system exhibited much stronger photocurrent compared to that of PS1 and TQ alone, indicating an enhanced charge transfer rate within the TQ/PS1 complex. Moreover, density functional theory (DFT) was conducted to calculate the highest-occupied molecular orbital (HOMO) and lowest-unoccupied molecular orbital (LUMO) energy levels of PS1, TQ, and CA. The LUMO level of PS1 was determined to be −2.61 eV, which lies between that of TQ (−3.31 eV) and CA (−0.26 eV) (Fig. 5e). This aligns well with the spectroscopic results that the fluorescence of PS1 can be quenched by TQ instead of CA (Fig. 5a and Supplementary Fig. 24). Furthermore, the electrostatic potential maps of PS1, TQ, CA were simulated by DFT calculations (Supplementary Fig. 26), with the red color representing the negative charge while the blue color indicating the positive charge. The adsorption energy of TQ/PS1 and CA/PS1 complexes were calculated to be −2526.45 kJ mol⁻¹ and −2452.46 kJ mol⁻¹, respectively (Supplementary

Table 1). Therefore, the negative adsorption energies indicate that both CA and TQ are prone to be spontaneously adsorbed by PS1, which is beneficial for in situ oxidation conversion of CA (CA/PS1) and electron transfer to TQ (TQ/PS1). Figure 5f presents the stable structure and electrostatic potential map of TQ/PS1 complex to visualize the electrostatic interactions between molecules after adsorption. There is an attractive electrostatic interaction between the regions of negative potential (red) with higher electron densities and the regions of positive potential (blue) with lower electron densities within the molecules. Electrons are more likely to move from the negative region to the positive region, facilitating electron transfer and thus promoting the Type I pathway.

Based on the above results, we propose a mechanism in which facilitated electron transfer promotes the Type I pathway of the TQ/PS1 complex. Specifically, within the CA/PS1 complex, CA undergoes oxidative conversion by the $^1O_2$ generated through PS upon light irradiation, resulting in the formation of TQ. Subsequently, an electron is transferred from the PS* to the adjacent electron-deficient molecule TQ. This transferred electron is then relayed from TQ to molecular oxygen, giving rise to the production of $O_2^{-\cdot}$. In essence, TQ functions as an electron transfer mediator to promote the Type I pathway. In this process, the oxygen required for the initial $^1O_2$ production can be regenerated from $O_2^{-\cdot}$ upon disproportionation reactions mediated by superoxide dismutase and Fenton/Haber–Weiss reaction. This conversion enables the original Type II PS to work effectively under hypoxic conditions.

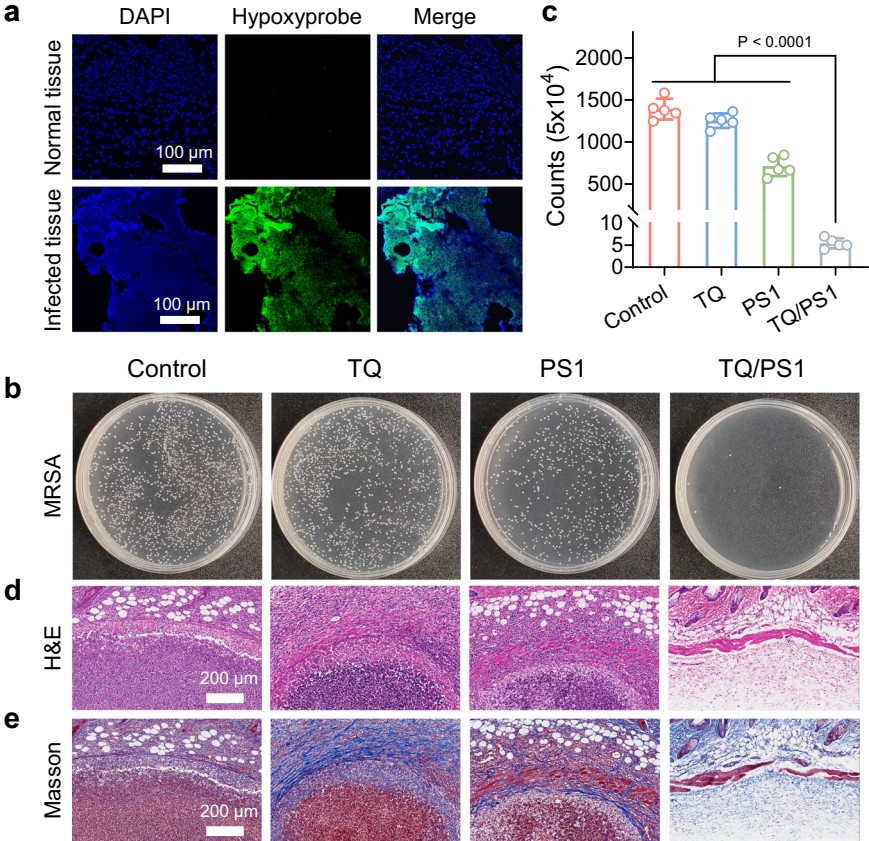

**Fig. 6 | In vivo bactericidal activities of TQ/PS1 complex. a** Detection of hypoxic conditions in both normal and infected tissues using the Hypoxyprobe Plus kit. Green fluorescence represents hypoxyprobe, blue fluorescence corresponds to DAPI. Scale bar = 100 μm. Data repeated independently ($n = 3$) with similar results. **b** Photographs of MRSA colonies in infected tissues treated with different groups upon light irradiation (60 mW/cm², 10 min) displayed on agar plates. **c** CFU counts of MRSA colonies from the antibacterial assay. Data presented as mean ± SD derived from $n = 5$ independent biological samples. Statistical significance was analyzed via one-way ANOVA test with a Tukey post hoc test. **d** H&E and **e** Masson-trichrome staining images of infected tissue slices at 7 d post-treatment in different groups. Scale bar = 200 μm. Data presented in (**d, e**) repeated independently ($n = 5$) with similar results. Different groups include TQ/PS1 (PS1 0.03 mg/kg, TQ 0.06 mg/kg), PS1 (0.03 mg/kg), TQ (0.06 mg/kg), and PBS.

To demonstrate the universality of facilitated Type I pathway mediated by TQ through enhanced electron transfer strategy, we applied two other Type II PSs, namely PS2 and PS3, and evaluated the $O_2^{-\cdot}$ generation of TQ/PS2 and TQ/PS3, respectively. Notably, both TQ/PS2 and TQ/PS3 significantly enhanced $O_2^{-\cdot}$ generation compared to PS2 and PS3 alone, which is similar to the case of TQ/PS1 (Supplementary Fig. 27). The fluorescence quenching of PS2 and PS3 in the presence of TQ was then performed, which showed a dose-dependent relationship. The Stern−Volmer plots exhibited a linear correlation between the concentration of TQ and $(I_0 − I)/I$ ratio, with $K_q$ values of $7.93 \times 10^9\,M^{-1}s^{-1}$ and $8.05 \times 10^9\,M^{-1}s^{-1}$, respectively (Supplementary Fig. 28). In addition, the fluorescence lifetimes of PS2 and PS3 were both shortened in the presence of TQ (Supplementary Fig. 29). Furthermore, the calculated LUMO level of PS2 (−2.13 eV) and PS3 (−2.60 eV) were higher than that of TQ (−3.31 eV), so that electron transfer between PS2 or PS3 and TQ could easily occur (Supplementary Fig. 30).

## In vivo antibacterial assessment

The in vivo experiments were subsequently conducted to evaluate the Type I PDT of the TQ/PS1 complex. An MRSA infection mouse model was established through subcutaneous injection of MRSA into the skin tissue of BALB/c mice. We examined local hypoxic conditions in infected tissues compared to normal tissues before antibacterial experiments. Hypoxyprobe, a specific tissue hypoxic probe with green fluorescence upon activation, was employed to visualize the hypoxic

conditions in bacterial-infected tissues[38,39]. As shown in Fig. 6a, the infected tissue exhibited bright green fluorescence in contrast to normal tissues, indicating the presence of a hypoxic microenvironment within the infected tissues. Then we evaluated the antibacterial efficiency of the TQ/PS1 complex. The infected mice were subcutaneously injected with TQ/PS1 complex, followed by light irradiation at 0.5 h post-administration (60 mW/cm², 10 min). The remaining bacteria after different treatments were visualized in agar plates as displayed in Fig. 6b, the bacteria were efficiently eliminated only upon treatment with TQ/PS1 complex under light irradiation. In addition, the bacterial survival rates in different groups were quantified through CFU counting (Fig. 6c). The antibacterial rate of TQ/PS1 complex was approximately 99.6%, whereas the antibacterial rate for PS1 alone was 48.8%, highlighting the efficient bactericidal ability of TQ/PS1 complex in vivo. The infected tissues were subsequently collected on day 7 for histological analysis. Hematoxylin and eosin (H&E) as well as Masson-trichrome staining revealed that the infected mice treated with TQ/PS1 complex had lower level of inflammatory cell infiltration and maintained a more normal skin tissue structure compared to the control groups (Fig. 6d, e). These results demonstrated that TQ/PS1 complex is an efficient antibacterial agent in the in vivo experiments.

In addition, the biosafety and biocompatibility evaluations were conducted both in vitro and in vivo. As shown in Supplementary Fig. 31, the cell viability of NIH 3T3 fibroblast cells remained above 80% under dark conditions but the damages to cells were observed upon light irradiation. The selectivity between cell and bacteria can be improved

by further nanofabrication (details see Supplementary Fig. 32). Then, the in vivo biocompatibility of TQ/PS1 complex was investigated. There were no significant changes in biochemical and hematological parameters between the mice treated with or without TQ/PS1 complex at day 1 and day 7 post-treatment (Supplementary Figs. 33 and 34). Moreover, histological analysis of major organs showed no obvious pathological changes (Supplementary Figs. 35 and 36) in TQ/PS1-treated group. In addition, skin infections often occur in open wounds, which are more sensitive to light-associated treatments. Thus, cytokine levels such as interleukin-6 (IL-6) and tumor necrosis factor-α (TNF-α), serving as markers of inflammation, were quantified at 24 h post-treatment from the wound tissue of mice to further evaluate the biocompatibility. These results indicated that there were no additional inflammatory side effects observed after treatment (Supplementary Fig. 37), indicating negligible in vivo toxicity of the TQ/PS1 complex.

## Discussion

In conclusion, we report that the combination of typical Type II PS and natural substrate CA enables a boosted Type I pathway with amplified generation of $O_2^{-\cdot}$. This coupled photodynamic agent can not only exert efficient bactericidal effects against *S. aureus* but also adapt to the hypoxic restriction suffered by traditional Type II PDT. Furthermore, the underlying mechanism is clearly revealed where CA serves as a $^1O_2$ activable electron acceptor precursor that generates TQ with an electron-deficient nature by PS under light irradiation, and TQ then acts as an electron transfer mediator which can receive electrons from PS and further transfer to nearby $O_2$ via Type I pathway, leading to the generation of $O_2^{-\cdot}$. Such a process is driven by matched redox potentials and spontaneous adsorption between PS and TQ. This work reveals a simple, effective, and universal method to promote Type I pathway and offers new insights into developing PSs for Type I PDT with great potential for translational research.

Collectively, TQ as an electron transfer mediator, facilitates the conversion of Type II PS into Type I by enhancing the electron transfer process. Notably, both CA and TQ are naturally occurring bioactive components commonly found in many medicinal plants and hold great potential for biomedical purposes with their affordability, safety, and effectiveness[40,41]. Furthermore, porphyrin derivatives have been one of the most successful PSs and received Food and Drug Administration (FDA) approval for treatment[42]. Thus, the magic fusion of traditional Type II PSs and phytochemicals without the influence on their origin structure, harnessing their intrinsic photophysical properties to enhance therapeutic performance, provides a new insight into more effective PDT. Together with their commercial availability, our facile approach holds implications for advancing the application of phytochemicals and the development of Type I PDT for translational applications.

## Methods

The research presented here complies with all relevant ethical regulations. All experiments involving animals were reviewed and approved by the guidelines of Guangdong Huawei Testing Co., Ltd. (approval number: 202303001), prior to commencing the study.

### Materials

All chemicals were obtained from commercial suppliers and used without further purification. Dimethylsulfoxide (DMSO), Tetrahydrofuran (THF), anhydrous N, N dimethylformamide (DMF), Lysogeny broth (LB), Chlorin e6 (Ce6, PS1), Protoporphyrin IX (PPIX, PS2), Tetrakis(4-carboxyphenyl)porphyrin (TCPP, PS3), 9,10-anthracene-diyl-bis(methylene) dimalonic acid (ABDA), 2′,7′-dichlorofluorescin diacetate (DCFH-DA), Dihydrorhodamine 123 (DHR 123) were purchased from Sigma-Aldrich. Vitamin C (VC), Carvacrol (CA), and thymoquinone (TQ) were purchased from Combi-Blocks. AnaeroPack was purchased from Mitsubishi Gas Chemical Company, Inc. LIVE/DEAD

viability stain kits were purchased from Invitrogen. Hypoxyprobe™-1 Plus Kit was purchased from Hypoxyprobe. ELISA kits were purchased from Shanghai Enzyme-linked Biotechnology Co., Ltd.

### Instruments

The PL lifetime measurements were carried out by Edinburgh Instruments FLS 1000. The photoluminescence (PL) spectra were characterized by the Perkin-Elmer LS 55 spectrometer and infinite M200 PRO microplate reader. The absorption of samples was performed on a Shimadzu UV-1700 ultraviolet-visible (UV–vis) spectrophotometer. Hydrodynamic size distributions were measured on a Malvern Instruments Zetasizer Nano ZS ZEN3600 analyzer at 25 °C. The scanning electron microscope (SEM) was recorded by JEOL JSM-7610F Plus. Confocal images were obtained by Leica SP8 confocal laser scanning microscope (CLSM). GC-MS analysis was conducted by Agilent Technologies 7890B GC system. Electrochemical experiments were conducted by ATUOLAB PGSTAT302N. Light irradiation was conducted using an L-150A-1 single optical fiber light source (Microscope X).

### Characterization of CA/PS1 complex formation

PS1 and CA were dissolved in DMSO with a concentration of 2 mg/mL and 40 mg/mL, respectively, as stock solution. Then, PS1/DMSO (20 μL) and CA/DMSO (37.5 μL) solution were mixed. The mixture was further mixed with distilled water (10 mL) to achieve a final concentration of 4 μg/mL of PS1 and 150 μg/mL of CA at room temperature to obtain CA/PS1 complex for further characterization by UV–Vis, SEM, and DLS.

### $^1O_2$ detection

ABDA (50 μM) was utilized as a $^1O_2$ indicator to detect the $^1O_2$ generation of chemicals. The cuvette with a mixture of ABDA and chemicals was irradiated with light (20 mW/cm$^2$) at various time, and the decrease of ABDA was monitored by the absorbance at 378 nm. As control, ABDA solution without chemicals was subjected to irradiation.

### $O_2^{-\cdot}$ detection

DHR 123 was used as the superoxide radical indicator. The cuvette with a mixture of DHR 123 and chemicals was subjected to light irradiation (20 mW/cm$^2$) at various time, and the fluorescence emission of DHR 123 at 500–600 nm was recorded after each irradiation. Before subjecting the above aqueous solution to light irradiation for the $O_2^{-\cdot}$ quenching experiment, 200 μM VC was introduced to the solution for pre-treatment purpose. As control, the DHR 123 solution without chemicals was subjected to irradiation.

### Total ROS detection

The ROS generation of chemicals was assessed by DCFH. DCFH solution was prepared by DCFH-DA hydrolysis in an alkaline solution. DCFH-DA solution (0.25 mL, 0.5 mg/mL) was added to NaOH solution (1 mL, 10 mM) for 30 min in the dark. And then, the formed solution was added to PBS (50 mM, 5 mL) to obtain the DCFH solution (40 μM). For ROS detection, DCFH solution (40 μM, 200 μL) was added into the solution obtained chemicals in PBS (800 μL). The mixture was subjected to light irradiation (5 mW/cm$^2$) at various times. The fluorescence emission of DCFH at 500–600 nm was recorded to obtain the ROS generation rate. As a control, the DCFH solution without chemicals was subjected to irradiation.

### Bacteria culture growth

Bacterial strains were purchased from ATCC. Bacterial cells were initially stored in −80 °C frozen tubes. They were taken out and incubated on LB agar plates at 37 °C overnight. Bacteria from single colonies were then transferred to a liquid LB culture medium and incubated in a MaxQ 4000 & Heidolph Incubator Shaker at 37 °C

overnight. The obtained bacterial cells were used for further characterization.

## Antibacterial activity in vitro

To assess the antibacterial activity, we tested the chemicals with liquid bacterial cultures. MRSA and *S. aureus* overnight cultures were diluted to approximately 0.1 $OD_{600}$ (optical density at 600 nm) and incubated with chemicals under normoxic or hypoxic conditions (The hypoxic conditions were maintained by using AnaeroPack ® with a matching culture container). After 4 h incubation, the phototoxicity of chemicals to bacteria was determined by light irradiation (60 mW/cm$^2$, 10 min). After irradiation, the bacteria were serially diluted tenfold, and each serial dilution was plated per dilution in triplicate onto LB agar. Finally, plates were incubated for 18–24 h at 37 °C to form bacteria colonies, and the plates were imaged and manually counted to determine the CFU of bacteria. For the VC quenching experiment, different concentrations of VC (0–500 μM) were added to the above bacterial suspensions with chemicals at the same time.

## Dead/live staining

The damages of bacterial envelop after incubation with chemicals before and after light irradiation were directly examined using the dead/live staining method. After incubation with chemicals, bacterial cells were subjected to light irradiation of 60 mW/cm$^2$ for 10 min. Then, PI and SYTO-9 were introduced to the bacterial suspension with a concentration of 10 μg mL$^{-1}$. The bacteria were washed with PBS and suspended in PBS after staining for 20 min. Then, the suspension was spotted on a microscope slide and examined using CLSM.

## Scanning electron microscope (SEM)

The morphologies of *S. aureus* cells incubated with chemicals before and after light irradiation were directly examined using SEM. After incubation with chemicals, bacterial cells were subjected to light irradiation of 60 mW/cm$^2$ for 10 min. Then, the samples were solidified by glutaraldehyde (2.5%) overnight and then washed (10 min every time) with gradient alcohol. Subsequently, the samples were processed by performing drop-coating of the sample solutions (5 μL) onto a silicon wafer. After drying the samples at room temperature, they were coated with gold for SEM image collection.

## Characterization of TQ/PS1 complex formation

PS1 and TQ were dissolved in DMSO with a concentration of 2 mg/mL as stock solution. Then, PS1/DMSO (20 μL) and TQ/DMSO (50 μL) solution were mixed. The mixture was further mixed with distilled water (10 mL) to achieve a final concentration of 4 μg/mL of PS1 and 10 μg/mL of TQ at room temperature to obtain TQ/PS1 complex for further characterization by UV–Vis, SEM, and DLS.

## Analysis of oxidation by-products of CA by GC-MS

PS1 and CA were mixed in THF solution (100 μg/mL of PS1 and 1 mg/mL of CA). The samples were subjected to light irradiation (60 mW/cm$^2$, 60 min), after which the mixture was applied for gas chromatography/mass spectrometry (GC-MS) analyses. The initial column temperature was set to 150 °C for 1 min and then increased at a rate of 10 °C per minute until it reached 250 °C. The temperature was maintained at 250 °C for 10 min. The injector port temperature was set at 250 °C, and helium was employed as the carrier gas at a flow rate of 1.0 mL/min[34].

## Fluorescence lifetime measurements

PS (10 μM) and TQ (10 mM) in DMF were placed in a long-neck quartz cell with closed rubber plug. Nanosecond fluorescence lifetime measurements were performed by using the FLS 1000 with excitation at 405 nm using EPL picosecond pulsed diode laser EPL-405.

## Measurement of photocurrent responses

The photocurrent responses experiment was performed by a three-electrode system. Carbon paper electrodes attached PS1, TQ, and TQ/PS1 were used as the working electrode, and the platinum electrode and the Ag/AgCl electrode were used as the counter electrode and reference electrode, respectively.

## Computational methods

The structures, energies, and electrostatic potential maps of molecules were calculated by Gaussian/16.C.02-AVX2, with B3LYP hybrid exchange-correlation functional. The Solvation Model Based on the Density (SMD) Model was used with DMSO as solvent.

## In vivo antibacterial activity of TQ/PS1 complex

Female BALB/c mice, aged 6–8 weeks, were employed for the study. 30 μL of MRSA ($1 \times 10^8$ CFU/mL) was subcutaneously injected into the dorsal skin of mice. After 1 day, bacterial-induced infections were established, and the infected regions were subcutaneously injected with 50 μL of PBS, PS1 (PS1 0.03 mg/kg), TQ (PS1 0.06 mg/kg), or TQ/PS1 complex (PS1 0.03 mg/kg, TQ 0.06 mg/kg), respectively. At 0.5 h post-injection, infected tissues of mice were treated with light irradiation (60 mW/cm$^2$) for 10 min. After 24 h, mice in different treatment groups were sacrificed. Infected skin tissues were harvested, homogenized using a tissue grinder, and the homogenized samples were spread on LB agar plates for colony-forming unit (CFU) counting. In addition, at 7 days post-treatment, infected skin tissues from various groups were harvested, fixed in 4% paraformaldehyde for 24 h, and then embedded and sectioned at a 5 μm thickness. The skin slices were stained with hematoxylin and eosin (H&E) or Masson-trichrome before imaging using an optical microscope.

## Staining of hypoxia tissue sections

The examination of hypoxia conditions in skin tissue was carried out according to the manufacturer's protocol. Briefly, bacterial-infected mice were subcutaneous injections with a Hypoxyprobe Plus kit. Then, the mice were sacrificed, and skin tissues were collected after 1 h of Hypoxyprobe Plus kit administration. The tissues were then fixed in 4% paraformaldehyde for 12 h, dehydrated in 40% sucrose/PBS, embedded in Optimal Cutting Temperature (OCT) compound (Tissue-Tek), and sectioned at a thickness of 8 μm. Tissues were then incubated with a specific secondary rabbit anti-FITC for 30 min. Finally, tissues were stained with DAPI for 20 min.

## In vitro and in vivo biosafety assessment

Briefly, 3T3 fibroblast cells were seeded on 96-well cell culture plates (concentration at $1.0 \times 10^4$ cells/well) and incubated for 24 h at 37 °C with 5% $CO_2$. Subsequently, they were treated with different concentrations of the TQ/PS1 complex. At 0.5 h post-incubation, the cells were subjected to light irradiation. Finally, the cytotoxicity assay was conducted based on an established methyl thiazolyl tetrazolium (MTT) assay. Biosafety evaluations in vivo were conducted using healthy BALB/c mice to assess the cytotoxicity of the TQ/PS1 complex. Mice were intravenously administered with 50 μL of TQ/PS1 complex (PS1 0.03 mg/kg, TQ 0.06 mg/kg). TQ/PS1 complex treated and untreated mice were sacrificed at 1 d and 7 d post-treatment for blood biochemistry analysis. Various organs were collected for subsequent histological examination. Besides, the biosafety assessment in open wounds model was conducted by creating a wound with a diameter of 1 cm on the back of the mice, with healthy mice set as blank control. The mice were then treated with PBS, TQ/PS1, and TQ/PS1 with light irradiation, respectively. The tissues were harvested at 1 day post-treatment and the cytokine levels in different collected tissues were measured using TNF-α ELISA kit (m1002095N) and IL-6 ELISA kit (ml098430) according to the manufacturer's instructions.

## Statistical analysis

All data are expressed as the mean ± standard deviation (SD). Inter-group and intra-group comparison analyses in each experiment were calculated by one- or two-way ANOVAs with a Tukey post hoc test. All statistical analyses were carried out by using Graphpad Prism (version 10.0.0) and Origin 2021. Probability ($P$) values < 0.05 were considered statistically significant. (*$P$ < 0.05, **$P$ < 0.01, ***$P$ < 0.001, and ****$P$ < 0.0001). ns means no significance.

## Reporting summary

Further information on research design is available in the Nature Portfolio Reporting Summary linked to this article.

## Data availability

The data generated in this study are available within the article, Supplementary Information, and Source Data. Source data are provided with this paper.

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

## Acknowledgements

This work was supported by the Singapore National Research Foundation (A-0009163-01-00, B.L.), National University of Singapore (E-467-00-0012-02, B.L.), and the National Natural Science Foundation of China (32271375, D.M.).

## Author contributions
B.L. supervised the project. J.H.Z., G.B.Q., and B.L. conceived and designed the experiments. H.Z. and X.H.Z. conducted theoretical calculations. D.M. and Y.C.F. conducted animal experiments. M.W., D.D.W., B.W.L., K.C.C, J.W.T., and Y.S. took part in the discussion and gave important suggestions. J.H.Z., G.B.Q., D. M., and B.L. co-wrote the paper.

## Competing interests
The authors declare no competing interests.
