## [Peer Review File · Nature Communications]

Reviewers' Comments:

Reviewer #1:

Remarks to the Author:

In this manuscript, the authors reported Thymoquinone as Electron Transfer Mediator to Convert Type II Photosensitizers to Type I. They designed the complex CA/PS1 nanoparticles by combination of Type II PS1 and natural substrate CA and suggested that CA could be transformed into thymoquinone (TQ) by local O_2 generated from the PS. The key point is that TQ could work as an efficient electron transfer mediator to enhance the Type I process for O_2 -• generation. Furthermore, they successfully applied this approach into another two classical Type II PSs (PS2 and PS3) and demonstrated the universality. This coupled photodynamic agent exhibited superior bactericidal activity even under hypoxic. The authors should reply to the following minor issues before considering accepting publication in nature communications.

1, The authors claimed in the manuscript that CA was converted into thymoquinone (TQ) by local O_2 generated from PS upon white light illumination. For therapeutic applications, it seems that carvacrol (CA) does not contribute anything here, but to some extent consumes a certain amount of O_2 . Why not directly chose TQ/PS complex for Type I PDT?

2, Benzoquinone derivatives such as p-benzoquinone have been reported in the literature as scavenging agents of O_2 -• for the reducing action (Small 2020, 16, 2001805, Small 2022, 18, 2104857), which seems to be somewhat contradictory to the role of electron transfer mediator proposed in this paper. How can the authors explain these opposite roles of benzoquinone derivatives?

3, In the main text about Fig. 5c, the words "delay fluorescence" was used by the authors. We wondered whether this "delay fluorescence" refers to the normal singlet excited state (S_1) radiation process or another radiation mechanism (such as P-type or E-type delay fluorescence et.al)?

4, Some differential analysis needs to be done, such as Figure 3.

5, Type I PDT show potentials for hypoxic tumor therapy, and the author also have mentioned the hypoxic bacterial-infected tissues. We are curious to know about the role of Type I PDT effect in the animal hypoxic inflammatory models. And why the authors did not design relevant animal models to verify the feasibility?

Reviewer #2:

Remarks to the Author:

The authors report in this manuscript that the combined use of white light irradiation and type II photosensitizers (PSs) can lead to more efficient elimination of bacteria in the presence of carvacrol (CA). This enhanced efficiency arises from the ability of CA to undergo conversion into thymoquinone (TQ) when exposed to light, resulting in a more effective Type I photodynamic therapy (PDT). However, it is worth noting that the conversion of CA to TQ and the transition from Type II to Type I PDT have been published, albeit with different light sources, which should be acknowledged and discussed in the paper. Moreover, while the study sheds some light on the more quantitative energy transfer of the PS/CA or PS/TQ, the specificity of such a modality to eliminate bacterial pathogens without causing harm to host cells is the most important criterion when assessing its clinical potential. Regrettably, the current study does not address this critical aspect, and no experiments have been conducted to evaluate potential detrimental effects on host cells. The manuscript lacks in vivo infection studies too.

In addition:

1. In the introduction, the authors mentioned: "Compared with Type II pathway, Type I PDT is less oxygen-dependent as O_2 -• is involved in the intracellular disproportionation reactions mediated by superoxide dismutase, which regenerates oxygen to alleviate hypoxia, and thus is more promising for antibacterial treatment under the hypoxic microenvironment." Although it is generally accepted that type I is less oxygen-dependent, the above explanation does not justify the claim. While it's

acknowledged that superoxide dismutase can regenerate oxygen from $O_2^{\bullet-}$, it concurrently consumes $O_2^{\bullet-}$, which is essential for the subsequent generation of other ROS, such as HO^{\bullet} . A more thorough explanation is needed to elucidate why, given this consumption, Type I PDT is still favored for antimicrobial treatments.

2. There are no statistical analyses of significance, sample sizes, or error bars for all relevant figures, which is unacceptable.

3. In the Antibacterial activity in vitro, the authors mentioned: "The hypoxic conditions were maintained by using AneroPack ® with a matching culture container." It is not clear, how the irradiation of bacterial culture was performed. Were the bacteria remained in the container, while white light irradiation was given? Had the oxygen level been measured? Was it hypoxia or anaerobic? Clarification ofw these and would be helpful.

4. It cannot be ruled out that antimicrobial activity with TQ/PS1 complex (as well as the CA/PS1 complex) under anaerobic conditions may be a Type III antimicrobial photoreaction, where oxygen is not required.

5. There is no description of the light source. While the use of white light is noted, there is no information on the distribution of wavelength. Although the use of white light may be convenient, the rationale behind not choosing a wavelength specific to PS1 needs to be clarified and justified.

6. TQ appears to be a photosensitizer, which has been well described in several studies, and well explains the enhanced antimicrobial activity of CA/PS1.

Jan 2nd, 2024

Dear reviewers,

We appreciate the reviewers' very helpful comments and revised the manuscript accordingly. A point-by-point reply has been included below. All the changes are marked with red color in the revised manuscript for your reference. Thank you very much for your kind attention.

Our responses to the reviewer's comments are summarized as follows.

COMMENTS TO AUTHOR:

Reviewer #1 (Remarks to the Author):

In this manuscript, the authors reported Thymoquinone as Electron Transfer Mediator to Convert Type II Photosensitizers to Type I. They designed the complex CA/PS1 nanoparticles by combination of Type II PS1 and natural substrate CA and suggested that CA could be transformed into thymoquinone (TQ) by local 1O_2 generated from the PS. The key point is that TQ could work as an efficient electron transfer mediator to enhance the Type I process for $O_2 \rightarrow \bullet$ generation. Furthermore, they successfully applied this approach into another two classical Type II PSs (PS2 and PS3) and demonstrated the universality. This coupled photodynamic agent exhibited superior bactericidal activity even under hypoxic. The authors should reply to the following minor issues before considering accepting publication in nature communications.

1, The authors claimed in the manuscript that CA was converted into thymoquinone (TQ) by local 1O_2 generated from PS upon white light illumination. For therapeutic applications, it seems that carvacrol (CA) does not contribute anything here, but to some extent consumes a certain amount of O_2 . Why not directly chose TQ/PS complex for Type I PDT?

Reply: We agree with the reviewer that directly applying TQ/PS1 complex for Type I PDT is more efficient than CA/PS1 complex, as it circumvents the conversion process from CA to TQ induced by singlet oxygen. During the revision, we have added the direct application of TQ/PS complex for Type I PDT.

In the earlier submission, CA/PS1 complex was initially found to enhance Type I ROS generation and therefore, antibacterial experiment under hypoxic conditions was conducted to assess the Type I PDT of the CA/PS1 complex. Afterwards, a series of experiments were performed to understand the mechanism behind Type I ROS overexpression of CA/PS1 complex. The results indicated that the intermediate TQ plays the role of electron transfer mediator to greatly boost Type I ROS generation for traditional Type II photosensitizers.

Based on the reviewer's advice to assess the Type I PDT of TQ/PS1 complex, we have conducted new experiments, and the results are added to the revised manuscript (Supplementary Fig. 22 and Fig. 4e). Meanwhile, Type I PDT of TQ/PS1 complex in the bacterial infection mouse model was also conducted and the result is provided as the answer to Question 5.

Supplementary Fig. 22. Photographs of the LB agar plates after the inoculation and overnight incubation of *S. aureus* cultures treated with TQ/PS1 (PS1 4 $\mu\text{g}/\text{mL}$, TQ 8 $\mu\text{g}/\text{mL}$), PS1 (4 $\mu\text{g}/\text{mL}$), and TQ (8 $\mu\text{g}/\text{mL}$) under hypoxic conditions with or without light irradiation (60 mW/cm^2 , 10 min).

Fig. 4e CFU counts of *S. aureus* in the presence or absence of light irradiation (60 mW/cm^2 , 10 min) treated with TQ/PS1 (PS1 4 $\mu\text{g}/\text{mL}$, TQ 8 $\mu\text{g}/\text{mL}$), PS1 (4 $\mu\text{g}/\text{mL}$), or TQ (8 $\mu\text{g}/\text{mL}$) under hypoxic conditions. Data presented as mean \pm SD derived from $n = 3$ independent samples. Statistical significance was analyzed via two-way ANOVA test with a Tukey post hoc test, **** $P < 0.0001$.

2, Benzoquinone derivatives such as p-benzoquinone have been reported in the literature as scavenging agents of $O_2^{\bullet-}$ for the reducing action (Small 2020, 16, 2001805, Small 2022, 18, 2104857), which seems to be somewhat contradictory to the role of electron transfer mediator proposed in this paper. How can the authors explain these opposite roles of benzoquinone derivatives?

Reply: We acknowledge that benzoquinone derivatives including TQ can quench $O_2^{\bullet-}$. The observed enhanced generation of $O_2^{\bullet-}$ in the presence of TQ/PS complex was due to close contact and matched LUMO levels (Fig. 5). To understand the phenomenon, we first detected $O_2^{\bullet-}$ generation when treated with TQ/PS1 at the operational concentration in both PBS solution (self-assembly) and DMF solution (well dispersed) to illustrate the importance of close proximity. As shown Figure R1, the signal of $O_2^{\bullet-}$ was greatly enhanced in the case of PBS solution; however, the signal of $O_2^{\bullet-}$ was quenched in DMF solution, which is an opposite phenomenon. These results revealed that close contact is the key factor for facilitating the electron transfer within TQ/PS complex to boost the generation of $O_2^{\bullet-}$.

Figure R1 Comparison of the PL intensity of DHR 123 treated with TQ/PS1 in PBS and DMF solution under light irradiation for $O_2^{\bullet-}$ detection. Data presented as mean \pm SD derived from $n = 3$ independent samples.

Furthermore, the detection of $O_2^{\bullet-}$ generation in DMF solution treated with different concentrations of TQ was conducted to further study the role of TQ in $O_2^{\bullet-}$ generation. As shown in Figure R2, the signal of $O_2^{\bullet-}$ in D and F decreased (14.24% for D, 10.17% for F) compared to B. Both PS1 and TQ are well dispersed (without close contact) in the case of DMF solution so the role of TQ as electron transfer mediator was greatly weakened and TQ began to work as quencher of $O_2^{\bullet-}$. When the concentration of TQ was further increased to 1 mg/mL, the signal of $O_2^{\bullet-}$ began to increase as presented in column H. The increased collisions between TQ and PS at high concentrations of TQ led to the facilitated Type I pathway mediated by TQ.

As a result, although TQ itself is a quencher of $O_2^{\bullet-}$, TQ also works as electron transfer mediator to facilitate Type I ROS generation when it meets the two critical requirements: close contact and matched LUMO levels.

Figure R2 Comparison of the PL intensity of DHR 123 alone and DHR 123 with PS1, TQ, or TQ/PS1 at different concentrations under light irradiation for $O_2^{\cdot-}$ detection in DMF solution. Data presented as mean \pm SD derived from $n = 3$ independent samples.

3, In the main text about Fig. 5c, the words “delay fluorescence” was used by the authors. We wondered whether this “delay fluorescence” refers to the normal singlet excited state (S1) radiation process or another radiation mechanism (such as P-type or E-type delay fluorescence et.al)?

Reply: We thank the reviewer for the valuable comments. It is the normal singlet excited state (S1) radiation process. For better understanding, “delayed fluorescence decay curve” has been changed to “fluorescence decay curve” in the revised manuscript.

4, Some differential analysis needs to be done, such as Figure 3.

Reply: We have added differential analysis to relevant figures according to the reviewer’s suggestion as follows:

Fig. 3c-d CFU counts from the antibacterial assay shown in a and b, respectively. Data in (c-d) are presented as mean \pm SD derived from $n = 3$ independent biological samples. Statistical significance was analyzed via two-way ANOVA test with a Tukey post hoc test, **** $P < 0.0001$.

5, Type I PDT show potentials for hypoxic tumor therapy, and the author also have mentioned the hypoxic bacterial-infected tissues. We are curious to know about the role of Type I PDT effect in the animal hypoxic inflammatory models. And why the authors did not design relevant animal models to verify the feasibility?

Reply: We appreciate the valuable suggestion from the reviewer. The relevant animal experiments were not conducted in the original submission because we focused on how type II photosensitizers can be converted to Type I and its underlying mechanism study.

During the revision, we conducted the related *in vivo* experiments to assess the therapeutic efficacy of the proposed method in treating animal models infected with bacteria. The analysis of the *in vivo* results was added to the revised manuscript and the discussion is shown as follows:

***In vivo* antibacterial assessment.** The *in vivo* experiments were subsequently conducted to evaluate the Type I PDT of the TQ/PS1 complex. An MRSA infection mouse model was established through subcutaneous injection of MRSA into the skin tissue of BALB/c mice. We examined local hypoxic conditions in infected tissues compared to normal tissues before antibacterial experiments. Hypoxyprobe, a specific tissue hypoxic probe with green fluorescence upon activation, was employed to visualize the hypoxic conditions in bacterial-infected tissues (*Nat. Nanotechnol.* 2016, 11, 941–947; *J. Invest. Dermatol.* 2008, 128, 1964–1968). As shown in Fig. 6a, the infected tissue exhibited bright green fluorescence in contrast to normal tissues, indicating the presence of a hypoxic microenvironment within the infected tissues. Then we evaluated the antibacterial efficiency of the TQ/PS1 complex. The infected mice were subcutaneously injected with TQ/PS1 complex, followed by light irradiation at 0.5 h post-administration (60 mW/cm², 10 min). The remaining bacteria after different treatments were visualized in agar plates as displayed in Fig. 6b, the bacteria were efficiently eliminated only upon treatment with TQ/PS1 complex under light irradiation. Additionally, the bacterial survival rates in different groups were quantified through CFU counting (Fig. 6c). The antibacterial rate of TQ/PS1 complex was approximately 99.6%, whereas the antibacterial rate for PS1 alone was 48.8%, highlighting the efficient bactericidal ability of TQ/PS1 complex *in vivo*.

The infected tissues were subsequently collected on day 7 for histological analysis. Hematoxylin and eosin (H&E) as well as Masson-trichrome staining revealed that the infected mice treated with TQ/PS1 complex had lower level of inflammatory cell infiltration and maintained a more normal skin tissue structure compared to the control groups (Fig. 6d-e). These results demonstrate that TQ/PS1 complex is an efficient antibacterial agent *in vivo*. Additionally, the biosafety and biocompatibility evaluations were conducted *in vivo*. As shown in Supplementary Fig. 31, there were no significant changes in biochemical and hematological parameters between the mice treated with or without TQ/PS1 complex. Moreover, histological analysis of major organs showed no obvious pathological changes (Supplementary Fig. 32) in TQ/PS1-treated group, indicating negligible *in vivo* toxicity of the TQ/PS1 complex.

Fig. 6 *In vivo* bactericidal activities of TQ/PS1 complex. (a) Detection of hypoxic conditions in both normal and infected tissues using the Hypoxyprobe Plus kit. Green fluorescence represents Hypoxyprobe, blue fluorescence corresponds to DAPI. Scale bar = 100 μm . (b) Photographs of MRSA colonies in infected tissues treated with different groups upon light irradiation (60 mW/cm^2 , 10 min) displayed on agar plates. (c) CFU counts of MRSA colonies from the antibacterial assay. Data presented as mean \pm SD derived from $n = 5$ independent biological samples. Statistical significance was analyzed via one-way ANOVA test with a Tukey post hoc test, **** $P < 0.0001$. (d) H&E and (e) Masson-trichrome staining images of infected tissue slices at 7 d post-treatment in different groups. Scale bar = 200 μm . Different groups include TQ/PS1 (PS1 0.03 mg/kg, TQ 0.06 mg/kg), PS1 (0.03 mg/kg), TQ (0.06 mg/kg), and PBS.

Supplementary Fig. 31 Hematology and blood biochemistry test for mice treated with TQ/PS1 complex at 7 d, with untreated mice serving as the control group. Biochemical and hematological parameters include alanine aminotransferase (ALT), aspartate aminotransferase (AST), creatinine (CREA), urea (UREA), white blood cell count (WBC), hemoglobin (HGB), hematocrit (HCT), platelet count (PLT), and the percentages of immune cells, specifically lymphocytes, monocytes, and neutrophil granulocytes. Data presented as mean \pm SD derived from $n = 3$ independent biological samples.

Supplementary Fig. 32 Histological images of major organs, including heart, liver, spleen, lung, and kidney, at 7 d after mice treated with TQ/PS1 complex, with untreated mice serving as the control group. Scale bar = 100 μ m

Reviewer #2 (Remarks to the Author):

The authors report in this manuscript that the combined use of white light irradiation and type II photosensitizers (PSs) can lead to more efficient elimination of bacteria in the presence of carvacrol (CA). This enhanced efficiency arises from the ability of CA to undergo conversion into thymoquinone (TQ) when exposed to light, resulting in a more effective Type I photodynamic therapy (PDT).

However, it is worth noting that the conversion of CA to TQ and the transition from Type II to Type I PDT have been published, albeit with different light sources, which should be acknowledged and discussed in the paper.

Reply: We appreciate the valuable suggestions provided by the reviewers. We focused on how type II photosensitizers can be converted to Type I and its underlying mechanism, facilitating the traditional Type II PSs to work under hypoxic conditions. According to the reviewer's suggestions, the discussion regarding CA to TQ and Type II to Type I PDT has been refined and added to the revised manuscript as follows:

1. CA to TQ:

(page 2) Similar to the previous report (*Sci. Transl. Med.* 2021, 13, eaba3571), we demonstrated that CA as a $^1\text{O}_2$ susceptible bioactive molecule can be *in situ* activated and then converted into thymoquinone (TQ) by local $^1\text{O}_2$ generated from PS upon light excitation (Fig. 1a).

(page 6) Indeed, it has been reported that the phenolic hydroxyl group of CA can undergo oxidation to form TQ with para-benzoquinone structure in the presence of $^1\text{O}_2$ (*Sci. Transl. Med.* 2021, 13, eaba3571; *Phys. Chem. Chem. Phys.* 2019, 21, 171–183; *Commun. Biol.* 2021, 4, 408).

It is important to note that the previous work reported the conversion of CA (a pre-photosensitizer) to TQ (a blue light sensitized photosensitizer) (*Sci. Transl. Med.* 2021, 13, eaba3571). However, our study revealed the electron transfer mechanism of TQ to facilitate the Type I ROS generation from Type II photosensitizers with successful demonstration of Type I PDT towards bacterial infection. It is important to note that **in our work, TQ does not function as a photosensitizer (no ROS production of TQ was observed, Figure R3) because the light source does not match the absorbance of TQ.**

Figure R3 DHR 123 and ABDA as $O_2^{\cdot-}$ and 1O_2 detector for the evaluation of ROS generation of TQ under light irradiation.

2. Type II to Type I PDT:

(page 2) In recent years, several strategies have been reported to design Type I PSs or endow Type II PSs with extra Type I ROS generation through cationization, heavy-atom regulation, and biotinylation of PSs (*ACS Nano* 2022, 16, 9130-9141; *Adv. Mater.* 2022, 34, 2108146; *Nat. Commun.* 2022, 13, 2225)

Moreover, while the study sheds some light on the more quantitative energy transfer of the PS/CA or PS/TQ, the specificity of such a modality to eliminate bacterial pathogens without causing harm to host cells is the most important criterion when assessing its clinical potential. Regrettably, the current study does not address this critical aspect, and no experiments have been conducted to evaluate potential detrimental effects on host cells.

Reply: We thank the reviewer for the valuable suggestions. The cytotoxicity of TQ/PS1 complex was evaluated both *in vitro* and *in vivo*. NIH/3T3 fibroblast cell (murine cell line) was chosen as a model cell for *in vitro* experiment because mouse was used for *in vivo* experiments. The results showed that TQ/PS1 complex did not exhibit obvious dark cytotoxicity (>80% viability) after being incubated with TQ/PS1 complex for 1 d (Figure R4).

Figure R4 Cell viability of NIH 3T3 fibroblast cells after being treated with different concentrations of TQ/PS1 complex for 24 h. The cell viability was measured by CCK-8 kit. Data presented as mean \pm SD derived from $n = 4$ independent biological samples. Statistical significance was analyzed via one-way ANOVA test with a Tukey post hoc test, $**P < 0.01$. ns, no significance.

We agree with the reviewer that the selective elimination of pathogens without harm to host cells is important. The selectivity can be obtained by post-modifications or nano-formulation, which endows the selectivity between bacteria and cells (*Chem. Soc. Rev.* 2019, 48, 415-427; *Nat. Commun.* 2022, 13, 3875). In the previous studies, we have also developed several strategies for selective ablating bacteria inside of host cells (*Angew. Chem. Int. Ed.* 2019, 58, 16229; *Adv. Mater.* 2020, 32, 2005222.)

During the revision, the *in vivo* evaluation of biosafety and biocompatibility of the TQ/PS1 complex was conducted. As shown in Supplementary Fig. 31, there were no significant changes between the mice treated with or without TQ/PS1 in terms of biochemical and hematological parameters. In addition, the major organs of the mice showed no obvious pathological changes based on the histological analysis results (Supplementary Fig. 32), indicating negligible *in vivo* toxicity of the TQ/PS1 complex.

In addition, the H&E and Masson staining of the infected tissues after different treatments were presented in Fig. 6d-e. The infected mice treated with TQ/PS1 complex had lower level of inflammatory cell infiltration and maintained a more normal skin tissue structure compared to all other control groups. Based on the above *in vivo* data, this approach can effectively eliminate pathogens, promoting tissue healing, while also possessing negligible *in vivo* toxicity.

Supplementary Fig. 31 Hematology and blood biochemistry test for mice treated with TQ/PS1 complex at 7 d, with untreated mice serving as the control group. Biochemical and hematological parameters include alanine aminotransferase (ALT), aspartate aminotransferase (AST), creatinine (CREA), urea (UREA), white blood cell count (WBC), hemoglobin (HGB), hematocrit (HCT), platelet count (PLT), and the percentages of immune cells, specifically lymphocytes, monocytes, and neutrophil granulocytes. Data presented as mean \pm SD derived from $n = 3$ independent biological samples.

Supplementary Fig. 32 Histological images of major organs, including heart, liver, spleen, lung, and kidney, at 7 d after mice treated with TQ/PS1 complex, with untreated mice serving as the control group. Scale bar = 100 μ m

Fig. 6d-e H&E and Masson-trichrome staining images of infected tissue slices at 7 d post-treatment in different groups. Scale bar = 200 μm . Different groups include TQ/PS1 (PS1 0.03 mg/kg, TQ 0.06 mg/kg), PS1 (0.03 mg/kg), TQ (0.06 mg/kg), and PBS.

The manuscript lacks *in vivo* infection studies too.

Reply: We appreciate the valuable suggestions from the reviewer. A bacterial-infected mouse model was built to evaluate the Type I PDT of the proposed method during the revision. The analysis of the *in vivo* results was added to the revised manuscript and the discussion is shown as follows:

***In vivo* antibacterial assessment.** The *in vivo* experiments were subsequently conducted to evaluate the Type I PDT of the TQ/PS1 complex. An MRSA infection mouse model was established through subcutaneous injection of MRSA into the skin tissue of BALB/c mice. We examined local hypoxic conditions in infected tissues compared to normal tissues before antibacterial experiments. Hypoxyprobe, a specific tissue hypoxic probe with green fluorescence upon activation, was employed to visualize the hypoxic conditions in bacterial-infected tissues (*Nat. Nanotechnol.* 2016, 11, 941–947; *J. Invest. Dermatol.* 2008, 128, 1964–1968). As shown in Fig. 6a, the infected tissue exhibited bright green fluorescence in contrast to normal tissues, indicating the presence of a hypoxic microenvironment within the infected tissues. Then we evaluated the antibacterial efficiency of the TQ/PS1 complex. The infected mice were subcutaneously injected with TQ/PS1 complex, followed by light irradiation at 0.5 h post-administration (60 mW/cm², 10 min). The remaining bacteria after different treatments were visualized in agar plates as displayed in Fig. 6b, the bacteria were efficiently eliminated only upon treatment with TQ/PS1 complex under light irradiation. Additionally, the bacterial survival rates in different groups were quantified through CFU counting (Fig. 6c). The antibacterial rate of TQ/PS1 complex was approximately 99.6%, whereas the antibacterial rate for PS1 alone was 48.8%, highlighting the efficient bactericidal ability of TQ/PS1 complex *in vivo*.

The infected tissues were subsequently collected on day 7 for histological analysis. Hematoxylin and eosin (H&E) as well as Masson-trichrome staining revealed that the infected mice treated with TQ/PS1 complex had lower level of inflammatory cell

infiltration and maintained a more normal skin tissue structure compared to the control groups (Fig. 6d-e). These results demonstrate that TQ/PS1 complex is an efficient antibacterial agent *in vivo*. Additionally, the biosafety and biocompatibility evaluations were conducted *in vivo*. As shown in Supplementary Fig. 31, there were no significant changes in biochemical and hematological parameters between the mice treated with or without TQ/PS1 complex. Moreover, histological analysis of major organs showed no obvious pathological changes (Supplementary Fig. 32) in TQ/PS1-treated group, indicating negligible *in vivo* toxicity of the TQ/PS1 complex.

Fig. 6 *In vivo* bactericidal activities of TQ/PS1 complex. (a) Detection of hypoxic conditions in both normal and infected tissues using the Hypoxyprobe Plus kit. Green fluorescence represents Hypoxyprobe, blue fluorescence corresponds to DAPI. Scale bar = 100 μ m. (b) Photographs of MRSA colonies in infected tissues treated with different groups upon light irradiation (60 mW/cm², 10 min) displayed on agar plates. (c) CFU counts of MRSA colonies from the antibacterial assay. Data presented as mean \pm SD derived from n = 5 independent biological samples. Statistical significance was analyzed via one-way ANOVA test with a Tukey post hoc test, ****P < 0.0001. (d) H&E and (e) Masson-trichrome staining images of infected tissue slices at 7 d post-treatment in different groups. Scale

bar = 200 μm . Different groups include TQ/PS1 (PS1 0.03 mg/kg, TQ 0.06 mg/kg), PS1 (0.03 mg/kg), TQ (0.06 mg/kg), and PBS.

Supplementary Fig. 31 Hematology and blood biochemistry test for mice treated with TQ/PS1 complex at 7 d, with untreated mice serving as the control group. Biochemical and hematological parameters include alanine aminotransferase (ALT), aspartate aminotransferase (AST), creatinine (CREA), urea (UREA), white blood cell count (WBC), hemoglobin (HGB), hematocrit (HCT), platelet count (PLT), and the percentages of immune cells, specifically lymphocytes, monocytes, and neutrophil granulocytes. Data presented as mean \pm SD derived from $n = 3$ independent biological samples.

Supplementary Fig. 32 Histological images of major organs, including heart, liver, spleen, lung, and kidney, at 7 d after mice treated with TQ/PS1 complex, with untreated mice serving as the control group. Scale bar = 100 μm .

In addition:

1. In the introduction, the authors mentioned: “Compared with Type II pathway, Type I PDT is less oxygen-dependent as $\text{O}_2^{\cdot-}$ is involved in the intracellular disproportionation reactions mediated by superoxide dismutase, which regenerates oxygen to alleviate hypoxia, and thus is more promising for antibacterial treatment under the hypoxic microenvironment.” Although it is generally accepted that type I is less oxygen-dependent, the above explanation does not justify the claim. While it’s acknowledged that superoxide dismutase can regenerate oxygen from $\text{O}_2^{\cdot-}$, it concurrently consumes $\text{O}_2^{\cdot-}$, which is essential for the subsequent generation of other ROS, such as $\text{HO}\cdot$. A more thorough explanation is needed to elucidate why, given this consumption, Type I PDT is still favored for antimicrobial treatments.

Reply: We thank the reviewer for the valuable suggestions. The catalytic cascades involving $\text{O}_2^{\cdot-}$ include disproportionation reaction, Fenton reaction, and Haber-Weiss reaction (*J. Am. Chem. Soc.* 2018, 140, 44, 14851–14859):

As mentioned by the reviewer, $\text{O}_2^{\cdot-}$ is also an essential raw material for the subsequent Fenton/Haber-Weiss reactions to generate other ROS species. However, it is worth noting that $\text{O}_2^{\cdot-}$ can also be catalyzed to O_2 for compensatory regeneration of O_2 in both Fenton reaction and Haber-Weiss reaction (*Acc. Chem. Res.* 2022, 55, 22, 3253–3264; *Small* 2021, 17, 2006742). Thus, oxygen can be widely regenerated in these reactions to compensate for oxygen depletion.

We have revised the discussion of type I PDT to be: “In contrast to the Type II pathway, Type I PDT demonstrates reduced reliance on oxygen, facilitated by the involvement of $\text{O}_2^{\cdot-}$ species in disproportionation reaction, along with the Fenton and Haber-Weiss reactions. These processes not only regenerate oxygen to alleviate hypoxia but also

stimulate the formation of other highly toxic ROS species. As a result, Type I PDT shows great potential for antibacterial treatment within hypoxic microenvironment.” in the revised manuscript.

2. There are no statistical analyses of significance, sample sizes, or error bars for all relevant figures, which is unacceptable.

Reply: According to the reviewer's suggestions, we have repeated the experiments and included the statistical analyses (i.e., significance and error bars) and a description of sample sizes for relevant figures.

3. In the Antibacterial activity in vitro, the authors mentioned: “The hypoxic conditions were maintained by using AnaeroPack[®] with a matching culture container.” It is not clear, how the irradiation of bacterial culture was performed. Were the bacteria remained in the container, while white light irradiation was given? Had the oxygen level been measured? Was it hypoxia or anaerobic? Clarification of these and would be helpful.

Reply: We thank the reviewer for the valuable suggestions to make the hypoxic protocol clearer.

1. The bacteria remained inside the container upon light irradiation.
2. We measured the oxygen level with a Portable Dissolved Oxygen (DO) Meter JPB1-610L in real time. Therefore, the oxygen level was calculated as follows:

$$\text{Oxygen level (\%)} = \frac{DO_2}{DO_1} * 21 \quad (1)$$

Where DO₁ and DO₂ refer to DO level before and 4 h after loading the AnaeroPack[®]. DO₁ and DO₂ were determined to be 8.07 and 0.78 as shown in Figure R5a and Figure R5c.

$$\text{Oxygen level (\%)} = \frac{0.78}{8.07} \times 21 \approx 2\%$$

Thus, it was hypoxia. Notably, the oxygen level can quickly approach 2% at around 30 min as shown in Figure R5b.

Figure R5 (a) DO level before loading the AnaeroPack[®]. (b) DO level 30 min after loading the AnaeroPack[®]. (c) DO level 4 h after loading the AnaeroPack[®].

4. It cannot be ruled out that antimicrobial activity with TQ/PS1 complex (as well as the CA/PS1 complex) under anaerobic conditions may be a Type III antimicrobial photoreaction, where oxygen is not required.

Reply: We thank the reviewer for the valuable suggestion. We created an anaerobic condition to exam if TQ/PS1 complex (as well as the CA/PS1 complex) can eliminate bacteria without oxygen (Figure R6). The oxygen level can quickly decrease to around 2% (Figure R6b) in 15 min and further drop to 0% (Figure R6c) at around 40 min. Under anaerobic conditions, the antibacterial activities of TQ/PS1 and CA/PS1 were greatly compromised as shown in Figure R7, indicating that TQ/PS1 and CA/PS1 cannot work under anaerobic conditions. Therefore, the antibacterial activity of TQ/PS1 complex (as well as the CA/PS1 complex) is not in a Type III mode.

Figure R6 (a) DO level before loading the AnaeroPack[®]. (b) DO level 15 min after loading the AnaeroPack[®]. (c) DO level 40 min after loading the AnaeroPack[®].

Figure R7 Photographs and CFU counts of *S. aureus* colonies from the antibacterial assay treated with TQ/PS1, CA/PS1, and the corresponding controls under anaerobic conditions with or without light irradiation (60 mW/cm², 10 min). TQ/PS1 (PS1 4 µg/mL, TQ 8 µg/mL), CA/PS1 (PS1 4 µg/mL, CA 150 µg/mL), PS1 (4 µg/mL), TQ (8 µg/mL), and CA (150 µg/mL). Data presented as mean ± SD derived from n = 3 independent biological samples. Statistical significance was analyzed via two-way ANOVA test with a Tukey post hoc test, *P < 0.05. ns, no significance.

5. There is no description of the light source. While the use of white light is noted, there is no information on the distribution of wavelength. Although the use of white light may be convenient, the rationale behind not choosing a wavelength specific to PS1 needs to be clarified and justified.

Reply: The model of the light source is “L-150A-1 single optical fiber light source, Microscope X”. Supplementary Fig.5 illustrates the spectrum of the light source, distributing continuously within the range of 400-700 nm, with a peak around 630 nm. The light source and information on wavelength distribution has been added to the revised manuscript.

Supplementary Figure 5. Wavelength distribution of the light source.

We chose to utilize this light source due to its convenience, versatility, and safety as compared to specific wavelength light source (*Nat. Commun.* 2022, 13, 2225; *Br. J. Dermatol.* 2008, 158, 740-746). Besides, based on the absorbance of PS1 as shown in Figure R8a, the wavelength distribution of light source and absorbance of PS1 match well (Figure R8b) so that PS1 can be excited well by our light source.

Figure. R8 (a) Normalized absorbance of PS1. (b) Overlaid spectra of light source and PS1 (450-750 nm).

6. TQ appears to be a photosensitizer, which has been well described in several studies and well explains the enhanced antimicrobial activity of CA/PS1.

Reply: We agree that TQ itself is a photosensitizer. However, the enhanced antimicrobial activity of CA/PS1 is not due to the photosensitive properties of TQ for the following reasons:

We first explain why TQ acts as a photosensitizer but did not induce cytotoxicity in our case. The absorbance of TQ primarily occurs within 400 nm and it is weak in the range of 400-700 nm as shown in Figure R9. Therefore, there are very limited overlaid ranges between the absorbance of TQ and the spectrum of our light source which mainly covers from 400 to 700 nm (inset of Figure R9). Besides, the molar absorption coefficient of TQ at 400 nm was calculated to be $36.08 \text{ M}^{-1} \text{ cm}^{-1}$ based on Figure R9 which is extremely low. Thus, TQ cannot be efficiently excited by our light source.

Figure R9 Absorbance of TQ (100 $\mu\text{g/mL}$). Inset is the overlaid spectra of light source and absorbance of TQ (300-800 nm).

Moreover, we included control groups consisting of TQ alone to evaluate the antimicrobial activity both *in vitro* and *in vivo*. TQ (8 $\mu\text{g}/\text{mL}$) itself did not exhibit obvious cytotoxicity to bacteria under light irradiation *in vitro* (Fig. 4d). Similarly, there is no significant difference after treatment by TQ alone (0.06 mg/kg) compared to the control group upon light irradiation *in vivo* (Fig. 6c).

Collectively, the enhanced antimicrobial activity of CA/PS1 is not due to the photosensitive properties of the generated TQ. Instead, TQ serves as an efficient electron transfer mediator to boost the electron transfer between PS and oxygen, promoting the conversion of O_2 to $\text{O}_2^{\cdot-}$ via electron transfer-based Type I pathway.

Fig. 4d CFU counts of *S. aureus* treated with TQ/PS1 (PS1 4 $\mu\text{g}/\text{mL}$, TQ 8 $\mu\text{g}/\text{mL}$), PS1 (4 $\mu\text{g}/\text{mL}$), and TQ (8 $\mu\text{g}/\text{mL}$) in the presence or absence of light irradiation (60 mW/cm², 10 min).

Fig. 6c CFU counts of MRSA colonies from the antibacterial assay. Data presented as mean \pm SD derived from $n = 5$ independent biological samples. Statistical significance was analyzed via one-way ANOVA test with a Tukey post hoc test, **** $P < 0.0001$.

College of Design and Engineering
Department of Chemical and Biomolecular Engineering

Thank you for your time, and we look forward to your final decision in due course.

Sincerely,

Prof. Bin Liu
Department of Chemical and Biomolecular Engineering
National University of Singapore
Singapore, 117585
Tel: 65-6516-8409
Fax: 65-6778-1936

Reviewers' Comments:

Reviewer #1:

Remarks to the Author:

The authors answered all my comments and questions very well by revising text and adding experiments. I suggest that this manuscript can be accepted by Nat. Comm.

Reviewer #2:

Remarks to the Author:

The paper is significantly improved with additional data and clarification. Unfortunately, the concerns about safety or specificity issues remain.

1. It is not clear why Figure R4 about the safety test with fibroblast in the rebuttal is not included in the paper. Was the cell viability assayed in the absence of light irradiation as stated as "dark cytotoxicity"? This is inadequate. Cell viability must be assessed with light irradiation, similar to the condition in the antibacterial assay. Significant cell viability loss was already seen at a high concentration of PS1/TQ, which can be worsened with light irradiation

2. The biocompatibility was evaluated in vivo on day 7 post-treatment, while bacterial load was measured at 24 hours post-treatment. This leaves a gap where acute toxicity or adverse effects of the treatment could have been missed. The study did not adequately address the specificity issue and somewhat misleading. For light irradiation, local side effects are typically assessed within 1 or 2 days after light irradiation. Moreover, skin infections occur predominantly in compromised skin such as open wounds or burn wounds, which are more sensitive to adverse factors of light-associated treatment and commonly used for evaluating biocompatibility.

3. The new in vitro and in vivo data should be updated in the abstract.

4. The killing efficiency is 99.6% in the paper and 99.99% in the abstract. Please make them consistent.

Apr 15, 2024

Dear reviewers,

We appreciate the reviewers' very helpful comments and revised the manuscript accordingly. A point-by-point reply has been included below. All the changes are marked with red color in the revised manuscript for your reference. Thank you very much for your kind attention.

Our responses to the reviewers' comments are summarized as follows.

REVIEWER COMMENTS

Reviewer #1 (Remarks to the Author):

The authors answered all my comments and questions very well by revising text and adding experiments. I suggest that this manuscript can be accepted by Nat. Comm.

Reply: We appreciate the strong support from the Reviewer #1.

Reviewer #2 (Remarks to the Author):

The paper is significantly improved with additional data and clarification. Unfortunately, the concerns about safety or specificity issues remain.

1. It is not clear why Figure R4 about the safety test with fibroblast in the rebuttal is not included in the paper. Was the cell viability assayed in the absence of light irradiation as stated as "dark cytotoxicity"? This is inadequate. Cell viability must be assessed with light irradiation, similar to the condition in the antibacterial assay. Significant cell viability loss was already seen at a high concentration of PS1/TQ, which can be worsened with light irradiation.

Reply: We appreciate the reviewer's insightful feedback. It is true that TQ/PS1 shows some dark toxicity, and the toxicity is even worse under light irradiation. The cell cytotoxicity is shown in Figure R1 and the Supplementary Figure 31.

Figure R1 (Supplementary Figure 31 in the revised manuscript). Cell viability of NIH 3T3 fibroblast cells treated with different concentrations of TQ/PS1 complex in the presence and absence of light irradiation (60 mW/cm², 10 min) using MTT assay. Data presented as mean ± SD derived from n = 4 independent biological samples.

This result is not surprising as the compound of TQ/PS1 does not have specificity to cells or bacteria. As the focus of the paper is to illustrate how Thymoquinone could serve as an electron transfer mediator to convert Type II photosensitizers to Type I for bacteria killing, we did not discuss the strategy to realize selectivity between cells and bacteria. It is well understood that the modality of PDT relies on the localized light activation of photosensitizers at the lesion site to generate highly reactive ROS for therapeutic purpose, offering unique spatiotemporal precision and selectivity due to the ability to direct illumination to the lesion (Nat. Rev. Clin. Oncol. 2020, 17, 657-674).

We understand the deep concern from the reviewer that specificity between cells and bacteria is the key component for any photosensitizer. Over the past few years, we and others have successfully demonstrated several strategies to realize high specificity between cells and bacteria through photosensitizer conjugation (Angew. Chem. Int. Ed. Engl. 2019, 59, 9288-9292; J. Med. Chem. 2023, 66, 14175–14187) or nanofabrication (Nat. Commun. 2019, 10, 4057). We therefore rely on external recognition element of photosensitizer probe or the surface properties of photosensitizer nanoparticles to offer the desired selectivity between cells and bacteria.

Herein, we use one strategy to demonstrate the selectivity between cells and bacteria by formulating PLGA-PEG nanoparticles with PS1 and TQ (abbreviated as TQ/PS1@PLGA), which has a diameter of about 200 nm measured using dynamic light scattering (Figure R2a). The fluorescence spectra of PS1 before and after the encapsulation are shown in Figure R2b. As shown in Figure R2c, the cellular uptake of TQ/PS1@PLGA was significantly reduced compared to that of TQ/PS1 at the same concentration. Due to reduced cellular uptake of TQ/PS1@PLGA, its toxicity is significantly reduced. Here we chose 15 minutes and 30 minutes for comparison to match the time used for *in vivo* experiments. At the same time scale and photosensitizer nanoparticle concentration, TQ/PS1@PLGA still

exhibited potent antibacterial activity while the survival rate of fibroblast cells was almost unaffected, indicating its good selectivity in killing bacteria, but not cells. These results indicate that our proposed photosensitizer system can be further optimized to achieve better therapeutic effects.

Figure R2. (a) Diameter of TQ/PS1@PLGA measured by DLS. (b) Fluorescence of TQ/PS1@PLGA and PS1. (c) Fluorescence imaging comparison of cellular uptake of TQ/PS1 and TQ/PS1@PLGA within 15 and 30 min by NIH 3T3 fibroblast cells. (d) Cell viability of NIH 3T3 fibroblast cells treated with different concentrations of TQ/PS1@PLGA in the presence and absence of light irradiation (60 mW/cm², 10 min) using MTT assay, n = 3. (e) Photographs of the LB agar plates after the inoculation and overnight incubation of *S. aureus* cultures treated with different concentrations of TQ/PS1@PLGA in the presence and absence of light irradiation (60 mW/cm², 10 min).

2. The biocompatibility was evaluated *in vivo* on day 7 post-treatment, while bacterial load was measured at 24 hours post-treatment. This leaves a gap where acute toxicity or adverse effects of the treatment could have been missed. The study did not adequately address the specificity issue and somewhat misleading. For light irradiation, local side effects are typically assessed within 1 or 2 days after light irradiation. Moreover, skin infections occur predominantly in compromised skin such as open wounds or burn wounds, which are more sensitive to adverse factors of light-associated treatment and commonly used for evaluating biocompatibility.

Reply: We thank the reviewer for the valuable comments. We have conducted the *in vivo* experiments and collected the data to evaluate the potential acute toxicity at 24 hours post-treatment. The data of blood tests and H&E staining of major organs at 24 hours

post-treatment are added to the revised manuscript as shown in Supplementary Figures 33 and 35.

Additionally, to assess the side effects of light-associated treatments, levels of pro-inflammatory cytokines such as IL-6 and TNF- α were measured at 24 hours post-treatment from the open wound tissues of mice. As shown in Supplementary Figure 36, the wound tissue treated with PBS exhibited higher levels of inflammation compared to normal tissue. More importantly, the wound tissue treated with TQ/PS1 in the presence of light irradiation showed comparable IL-6 and TNF- α levels to those in the TQ/PS1 and PBS alone groups. This finding demonstrates that PDT does not induce additional inflammatory responses, thus confirming the good biosafety of this therapeutic approach.

Supplementary Figure 33. Hematology and blood biochemistry test for mice treated with TQ/PS1 complex at 1 d, with untreated mice serving as the control group. The untreated mice served as the control group. The analysis included biochemical indicators such as

alanine aminotransferase (ALT), aspartate aminotransferase (AST), creatinine (CREA), urea (UREA), white blood cell count (WBC), hemoglobin (HGB), hematocrit (HCT), platelet count (PLT), and the percentages of immune cells, specifically lymphocytes, monocytes, and neutrophil granulocytes. Data presented as mean \pm SD derived from $n = 3$ independent biological samples.

Supplementary Figure 35. Histopathology evaluation of major organs (heart, liver, spleen, lung, kidney) at 1 d after mice treated with TQ/PS1 complex. The untreated mice were set as the control group. Scale bar = 100 μ m.

Supplementary Figure 36. Secretion levels of IL-6 (a) and TNF- α (b) collected from tissues of open wounds were measured at 1 d post-treatment, with healthy mice serving as blank control. Data presented as mean \pm SD derived from $n = 3$ independent biological samples.

3. The new *in vitro* and *in vivo* data should be updated in the abstract.

Reply: We thank the reviewer for the valuable comments. A new paragraph has been added to the revised abstract to describe the new added data as follows: “The Type I PDT against *S. aureus* was demonstrated under hypoxic conditions *in vitro*. Furthermore, this coupled photodynamic agent exhibited significant bactericidal activity with an antibacterial rate of 99.6% for the bacterial-infection mice in the *in vivo* experiments”.

4. The killing efficiency is 99.6% in the paper and 99.99% in the abstract. Please make them consistent.

Reply: The value of killing efficiency in the abstract has been revised to keep it consistent with that in the manuscript.

Thank you for your time, and we look forward to your final decision in due course.

Sincerely,

Prof. Bin Liu
Department of Chemical and Biomolecular Engineering
National University of Singapore
Singapore, 117585
Tel: 65-6516-8409
Fax: 65-6778-1936

Reviewers' Comments:

Reviewer #2:

Remarks to the Author:

The authors provide convincing data addressing previous concerns about safety and selectivity. Especially, figure R2 in the rebuttal addressed selectivity at cellular levels although the corresponding study was not conducted in vivo. This Figure R2 should be added to the manuscript as it is important to demonstrate the potential selectivity.

May 15, 2024

Dear reviewers,

Our responses to the reviewers' comments are summarized as follows.

REVIEWER COMMENTS

Reviewer #2 (Remarks to the Author):

The authors provide convincing data addressing previous concerns about safety and selectivity. Especially, figure R2 in the rebuttal addressed selectivity at cellular levels although the corresponding study was not conducted in vivo. This Figure R2 should be added to the manuscript as it is important to demonstrate the potential selectivity.

Reply: We appreciate the valuable suggestions from the Reviewer #2. Figure R2 has been added into the revised manuscript as Supplementary Figure 32.

Thank you for your time, and we look forward to your final decision in due course.

Sincerely,

Prof. Bin Liu
Department of Chemical and Biomolecular Engineering
National University of Singapore
Singapore, 117585
Tel: 65-6516-8409
Fax: 65-6778-1936